# The impact of monosomies, trisomies and segmental aneuploidies on chromosomal stability

Dorine C. Hintzen[1]☯, Mar Soto[1]☯, Michael Schubert[1], Bjorn Bakker[2], Diana C. J. Spierings[2], Karoly Szuhai[3], Peter M. Lansdorp[2], Roel J. C. Kluin[4], Floris Foijer[2], René H. Medema[1]*, Jonne A. Raaijmakers[1]*

**1** Oncode Institute, Division of Cell Biology, The Netherlands Cancer Institute, Amsterdam, The Netherlands, **2** European Research Institute for the Biology of Ageing, University of Groningen, University Medical Center Groningen, Groningen, The Netherlands, **3** Leids Universitair Medisch Centrum, Leiden, The Netherlands, **4** Genomics Core Facility, The Netherlands Cancer Institute, Amsterdam, The Netherlands

☯ These authors contributed equally to this work.
* j.raaijmakers@nki.nl (JAR); r.medema@nki.nl (RHM)

**Data Availability Statement:** The CNV sequencing is deposited in the SRA with BioProject ID: PRJNA791572. The RNA sequencing data is deposited in the GEO data base with the ID:

## Abstract

Aneuploidy and chromosomal instability are both commonly found in cancer. Chromosomal instability leads to karyotype heterogeneity in tumors and is associated with therapy resistance, metastasis and poor prognosis. It has been hypothesized that aneuploidy *per se* is sufficient to drive CIN, however due to limited models and heterogenous results, it has remained controversial which aspects of aneuploidy can drive CIN. In this study we systematically tested the impact of different types of aneuploidies on the induction of CIN. We generated a plethora of isogenic aneuploid clones harboring whole chromosome or segmental aneuploidies in human p53-deficient RPE-1 cells. We observed increased segregation errors in cells harboring trisomies that strongly correlated to the number of gained genes. Strikingly, we found that clones harboring only monosomies do not induce a CIN phenotype. Finally, we found that an initial chromosome breakage event and subsequent fusion can instigate breakage-fusion-bridge cycles. By investigating the impact of monosomies, trisomies and segmental aneuploidies on chromosomal instability we further deciphered the complex relationship between aneuploidy and CIN.

## Introduction

Chromosomal instability (CIN) and aneuploidy are both acknowledged hallmarks of cancer. Although these terms are often used interchangeably, they have different implications and it is therefore crucial to stress the difference. While aneuploidy refers to the genomic status of a cell with an abnormal chromosome content at a given time, CIN refers to the behavior of cells that display dynamic chromosomal content due to persistent segregation errors [1, 2]. There are two main types of aneuploidy; numerical aneuploidy, referring to the gain or loss of whole chromosomes, and segmental aneuploidy, referring to the gain or loss of parts of

GSE193437. The single cell sequencing data was deposited to ENA with the ID: PRJEB49640.

**Funding:** Marie Curie Initial Training Network Project PLOIDYNET: Mar Soto, Rene H. Medema FP7-PEOPLE-2013; KWF Kankerbestrijding (DCS): Jonne A. Raaijmakers KWF- Young Investigator Grant- 12233. The funders had no role in study design, data collection and analysis, decision to publish, or preparation of the manuscript.

**Competing interests:** No, there is no conflict of interest. My manuscript contains the following statement: "The authors declare that they have no conflict of interest."

chromosomes. Besides gaining or losing genetic information, segmental aneuploidies also often result in structural chromosomal aberrations such as translocations and dicentric chromosome formation [3].

Aneuploidy has been shown to be detrimental in all organisms, leading to growth and developmental impairments [4, 5]. However, aneuploidy is very common in cancer with approximately 90% of all solid tumors harboring abnormal karyotypes [6]. Single cell analyses have shown that besides stable aneuploidies, cells often display karyotype variation within a single tumor, due to ongoing CIN. Ongoing CIN allows cells to widely sample a range of different genotypes that can eventually be selected to match the requirements faced under stress [7]. Indeed, continuous karyotype deviations and subsequent karyotype selection have been shown to confer adaptation to challenging environments in single-cell organisms [8–11] as well as in mammalian cell culture [12–14]. Moreover, several studies have demonstrated that ongoing segregation errors in tumors are associated with enhanced therapy resistance, poor prognosis, and metastasis, thus reflecting adaptation conferred by CIN [15, 16]. However, the oncogenic potential of CIN can differ per tissue and high levels can in some cases also decrease tumorigenic potential [17–20].

Due to the high prevalence of aneuploidy and CIN in cancer, much attention has been put over the last years to reveal the underlying causes. Some causes have been identified such as a weakened spindle checkpoint, defects in chromosome cohesion, abnormal kinetochore-microtubule attachments and supernumerary centrosomes [21–23]. Nevertheless, mutations in key regulators of these processes are rarely found in tumors [21], and thus the exact underlying cause for CIN remains unclear in the majority of cases. While aneuploidy is a consequence of CIN, it remains controversial if aneuploidy *per se* can be an underlying cause of CIN. Some studies have indeed suggested that deviating karyotypes can instigate instability [24–29]. For instance, CIN has been suggested to be the result of specific gains or losses of chromosomes that contain key regulators of proper segregation in mitosis [25, 30]. Furthermore, it has been shown that the presence of extra chromosomes disrupts proteostasis and saturation of the protein folding machinery can result in misfolded proteins, aggregate formation and activation of protein degradation pathways [31–34]. Subsequently, this can result in the deregulation of specific proteins that highly depend on the chaperone machinery. These proteins involve, among others, key players of replication, therefore expression changes possibly increase replication stress and lead to CIN [28]. Importantly, it has also been shown that not all whole chromosome imbalances induce CIN [24, 25, 35], indicating that the mere presence of additional chromosomes may not always be sufficient to induce chromosomal instability.

Here, we set out to investigate the impact of a large panel of *de novo* induced stable aneuploidies on chromosome stability in a near-diploid human cell line. Specifically, we aim to understand whether different types of aneuploidies have a different impact on CIN, as well as understand what features of aneuploidy drive instability. Importantly, while most studies have focused mostly on trisomies, in this study we also include aneuploid clones with only monosomies, segmental aneuploidies as well as more complex karyotypes.

## Materials and methods

### Cell culture, cell lines, and reagents

hTert-immortalized retinal pigment epithelium (RPE-1) cells were obtained from ATCC and RPE-1 p53kd cells were kindly provided by R. Beijersbergen. RPE-1 p53kd cells were generated by transduction with pRetroSuper-p53 (with the shRNA sequence 5′-CTACATGTGTAACA GTTCC-3′) and selected with Nutlin-3a for functional loss of p53. H2B-Dendra2 cells were made as described in [36]. Cells were cultured at 37C at 5% $CO_2$ in Advanced Dulbecco's

Modified Eagle Medium: Nutrient mixture F-12 (DMEM-F12) with Glutamax (GIBCO), supplemented with 12% FCS (Clontech), 100 U/ml penicillin (Invitrogen), 100 μg/ml streptomycin (Invitrogen) and 2mM UltraGlutamin (Lonza). Inhibitors were all dissolved in DMSO and were used at the following concentrations: GSK923295 50nM, NMS-P715 480nM, Nutlin-3a 10μM, MG132 75nM and 150nM, 17-AAG 15nM and 30nM, UMK57 100nM.

### Generating aneuploid clones

Clones were generated by blocking RPE-1 p53kd cells in Thymidine for 14 hours, after which they were released in medium containing a combination of an Mps1 inhibitor (NMS-P715) (480uM) and a CENP-E inhibitor (GSK923295) (50nM) for 8 hours to induce whole chromosome aneuploidies and segmental aneuploidies as shown in Soto et al., 2017. After treatment, cells were collected by trypsinization and cells were plated single cell in 384-well plates. On the same day, a total of 2688 wells were examined for the presence of individual cells to ensure a single cell was present. Of those, 481 wells contained a single cell. To be confident that a missegregation took place, only wells containing a single cell harboring a micronucleus were selected, which were 119 cells. Out of these, 17 established until a full clone, while others stopped proliferating. CNV sequencing showed an aneuploid karyotype for 13 clones. Thus, out of 481 single plated cells, we obtained 13 aneuploid clones, a success percentage of 3,5%. Live cell imaging of these clones showed that aneuploid clones tend to proliferate slower. To also generate clones that did not experience a MN during their generation, we selected established clones with a slow growth phenotype. With this approach we significantly improved the success rate to 17% (out of all single cell plated cells). These are the clones starting from WA8, and clone WA5.

The gain of 10q in parental RPE-1 cells, deriving from an imbalanced fusion of the q-arm of chromosomes 10 to the X chromosome ([18], ATCC) and chromosome 12, present in a fraction of RPE-1 cells [36, 37] were not considered *de novo* aneuploidies (S1 Fig).

### Live-cell imaging

For live-cell imaging, cells were grown in a Lab-Tek II chambered coverglass (Thermo Science). Images were acquired every 5 minutes using a DeltaVision Elite (Applied Precision) microscope maintained at 37˚C, 40% humidity and 5% CO2, using a 20x 0.75 NA lens (Olympus) and a Coolsnap HQ2 camera (Photometrics) with 2 times binning. Image analysis was done using ImageJ software. DNA was visualized using 0,25μM SiR-DNA (Spirochrome).

### Immunofluorescence

Cells were grown on 24-mm glass coverslips and fixed in 3.7% formaldehyde/0.5% Triton X-100 in PBS for 15 min at room temperature. Primary antibodies were incubated 1 hour at room temperature and secondary antibodies were incubated for 2h at room temperature, both dissolved in PBS 0,1% Tween. The following antibodies were used: Crest (CS1058, Cortex Biochem), Histon H3 phospho (06–570, Upstate). Secondary antibodies for immunofluorescence were Alexa Fluor 488, Alexa Fluor 568, and Alexa Fluor 647 (Molecular Probes). DAPI was added to all samples before mounting using Prolong Gold antifade (Invitrogen). Images were acquired on a DeltaVision Elite microscope (Applied Precision) with a PlanApo N 60×/NA 1.42 objective (Olympus) and a Coolsnap HQ2 camera (Photometrics).

### Cell growth analysis

Proliferation was measured by using the Incucyte FRL (Essen BioScience) or the Lionheart FX automated microscope (Biotek). For the Incucyte, 250 cells were plated in 96-well plates.

Three or four replicate wells were imaged per cell line (phase-contrast) with a 4 h interval for 6 d. Confluency was determined by IncuCyte FLR software 2011A Rev2 and IncuCyte Zoom software 2013B Rev1 using phase-contrast images, and doubling times were calculated using GraphPad Prism 6 software.

For the Lionheart, 500 cells were plated in 96-well plates. Two or three replicate wells were imaged per clone with a 4 h interval for 5 days, and cells were stained with the DNA dye siR-DNA (Spirochrome). Proliferation rates were measured by performing cell count analysis using Gen5 software (BioTek) and doubling times were calculated using GraphPad Prism 8 software.

## Copy number analysis

DNA was isolated using the DNeasy Blood and Tissue kit (Qiagen) according to the manufacturer's protocol. The number of double-stranded DNA in the genomic DNA samples was quantified by using the Qubit. dsDNA HS Assay Kit (Invitrogen, cat no Q32851). Up to 2000 ng of double-stranded genomic DNA was fragmented by Covaris shearing to obtain fragment sizes of 160–180 bp. Samples were purified using 1.8X Agencourt AMPure XP PCR Purification beads according to the manufacturer's instructions (Beckman Coulter, cat no A63881). The sheared DNA samples were quantified and qualified on a BioAnalyzer system using the DNA7500 assay kit (Agilent Technologies cat no. 5067–1506). With an input of maximum, 1 μg sheared DNA, library preparation for Illumina sequencing was performed using the KAPA HTP Library Preparation Kit (KAPA Biosystems, KK8234). During library enrichment, 4–6 PCR cycles were used to obtain enough yield for sequencing. After library preparation, the libraries were cleaned up using 1X AMPure XP beads. All DNA libraries were analyzed on a BioAnalyzer system using the DNA7500 chips for determining the molarity. Up to eleven uniquely indexed samples were mixed together by equimolar pooling, in a final concentration of 10nM, and subjected to sequencing on an Illumina HiSeq2500 machine in one lane of a single read 65 bp run, according to manufacturer's instructions.

Low-coverage whole-genome samples, sequenced single-end 65 base pairs on the HiSeq 2500 were aligned to GRCh38 with bwa version 0.7, mem algorithm [38]. The mappability per 15 kilobases on the genome, for a samples' reads, phred quality 37 and higher, was rated against a similarly obtained mappability for all known and tiled 65bp subsections of GRCh38; a reference genome based mappability provided by QDNAseq [39], using a GRCh38 lifted version (https://github.com/asntech/QDNAseq.hg38.git). QDNAseq segments data using an algorithm by DNAcopy [40] and calls copy number aberrations using CGHcall [41], and visualization was adapted from the QDNAseq code.

## Calculation of number of imbalanced/gained/lost genes

For clones harboring whole chromosome aneuploidies, the number of imbalanced/ gained / lost coding genes was calculated by determining the aneuploid chromosomes per clone using CNVseq data. Then per clone the number of coding genes located on the monosomic chromosomes were summed up to obtain number of lost coding genes, the number of coding genes located on the trisomic chromosomes were summed up to obtain number of gained coding genes and for the number of imbalanced genes, the total of lost and gained coding genes was determined. To determine the number of coding genes per chromosome we made use of Ensembl release 79 using the genome assembly GRCh38, only considering protein-coding genes. For segmental aneuploidies we determined the breakpoints using CNVseq data and calculated the number of coding genes on the gained, lost or imbalanced segments with the same method as described above.

## Single-cell whole genome sequencing (scWGS), data processing and analysis

Single cells were lysed and stained in a nuclei isolation buffer (100 mM Tris-HCl [pH 7.4], 150 mM NaCl, 1 mM CaCl2, 0.5 mM MgCl2, 0.1% NP-40, and 2% BSA). Nuclei were stained with propidium iodide and Hoechst 33258 at concentrations of 10 μg/mL. Individual nuclei of G1 cells were sorted directly into 5 μL freezing buffer (50% PBS, 7.5% DMSO, and 42.5% 2X Pro-Freeze-CDM [Lonza]) in 96-well plates using a FACSJazz cell sorter (BD Biosciences). Plates were spun down at 500g for 5 min at 4C prior to storage at -80C until library preparation. Pre-amplification free scWGS library preparation was performed as described previously [42]. Libraries were sequenced on an Illumina HiSeq2500 sequencing platform, with clusters generated using the cBot. Raw sequencing data were demultiplexed and converted into fastq format using standard Illumina software (bcl2fastq version 1.8.4). Indexed bam-files were generated by mapping to GRCh37 using bowtie2 (version 2.2.4). Duplicate reads were marked using BamUtil (version 1.0.3). Copy number variations were called using the R-package AneuFinder, and quality control was performed as described before [43]. The analysis was done using 1Mb bins.

## COBRA and BAC-FISH analysis

From RPE-1 p53kd parental and clone SA6, cells were harvested using a metaphase harvesting protocol described earlier [18]. Metaphase cells were further analyzed using a multicolor-FISH karyotyping technique called COBRA-FISH, allowing identification of chromosomes based on spectrally distinct colors, according to protocols described earlier [44]. BAC probes were labeled using nick translation and imaged as described in detail earlier [1]. A Leica DM-RXA epifluorescence microscope (Leica, Wetzlar, Germany) was used equipped with a 100-W mercury lamp and computer-controlled filter rotor with excitation and emission filters for visualization of DAPI, DEAC, fluorescein, lissamine, Cy5, Cy7 using Leica Block A, DEAC (Chroma Technology), HQ-FITC, Pinkel set plus SP 570, HQ-Cy5, and HQ-Cy7 filters, respectively. A 63X objective (N.A. 1.32 PL APO, Leica) was used.

## Immunoblotting

RPE-1 cells were harvested and lysed using Laemmli buffer (120 mM Tris, pH 6.8, 4% SDS, and 20% glycerol). Equal amounts of protein were separated on a polyacrylamide gel and subsequently transferred to nitrocellulose membranes. Membranes were probed with the following primary antibodies: LC3B (Rabbit, Sigma-Aldrich, L7543), alpha-Tubulin (mouse, Sigma, t5168). HRP-coupled secondary antibodies (Dako) were used in a 1:1000 dilution. The immunopositive bands were visualized using ECL Western blotting reagent (GE Healthcare) and a ChemiDoc MP System (Biorad).

## RNA sequencing and data analysis

RPE-1 p53KD cells (parental and clones) were harvested in buffer RLT (Qiagen). Strand-specific libraries were generated using the TruSeq PolyA Stranded mRNA sample preparation kit (Illumina). In brief, polyadenylated RNA was purified using oligo-dT beads. Following purification, the RNA was fragmented, random-primed and reverse transcribed using SuperScriptII Reverse Transcriptase (Invitrogen). The generated cDNA was 3′ end-adenylated and ligated to Illumina Paired-end sequencing adapters and amplified by PCR using HiSeq SR Cluster Kit v4 cBot (Illumina). Libraries were analyzed on a 2100 Bioanalyzer (Agilent) and subsequently

sequenced on a HiSeq2500 (Illumina). We performed RNAseq alignment using TopHat 2.1.1. on GRCh38 and counted reads using Rsubread 2.4.3 (Ensembl 102).

We calculated differential expression between two biological replicates of the parental and each clone, between the monosomic (WA1, WA2, WA5) and trisomic clones (WA8, WA11, WA12), as well as the Mps1 and CENP-E inhibitor treated vs control, using DESeq2 1.31.3. We tested for gene set differences by using a linear regression model of the Wald statistic (as reported by DESeq2) between genes belonging to a set vs. genes not belonging to a set. Gene set collections included MSigDB hallmarks (2020) and Gene Ontology (2021). For the UPR analysis, we chose genes annotated with GO categories GO:0036500 (ATF6-mediated unfolded protein response), GO:0036498 (IRE1-mediated unfolded protein response), and GO:0036499 (PERK-mediated unfolded protein response) or their child terms.

### RNA isolation and qRT-PCR analysis

Total RNA was extracted from untreated RPE-1 cells. RNA isolation was performed by using the Qiagen RNeasy kit and quantified using NanoDrop (Thermo Fisher Scientific). cDNA was synthesized using SuperScript III reverse transcription, oligo dT (Promega), and 1000 ng of total RNA according to the manufacturer's protocol. Primers were designed with a melting temperature close to 60 degrees to generate 90–120-bp amplicons, mostly spanning introns. cDNA was amplified for 40 cycles on a cycler (model CFX96; Bio-Rad Laboratories) using SYBR Green PCR Master Mix (Applied Biosystems). Target cDNA levels were analyzed by the comparative cycle (Ct) method and values were normalized against GADPH expression levels.

| Primer | Forward | Reverse |
| --- | --- | --- |
| Actin | GCCGATCCACACGGAGTACTT | TTGCCGACAGGATGCAGAA |
| IL1B | TGGCAATGAGGATGACTTGT | TCGGAGATTCGTAGCTGGAT |
| IL6 | ACTCACCTCTTCAGAACGAATTG | CCATCTTTGGAAGGTTCAGGTTG |
| CCL2 | TTGCTTGTCCAGGTGGTCCAT | AAGATCTCAGTGCAGAGGCTC |
| CXCL1 | AGTGTGAACGTGAAGTCCCC | GGGGATGCAGGATTGAGGC |
| CXCL3 | ACCTCAAGAACATCCAAAGTGTG | GATGCGGGGTTGAGACAAG |

### Growth assay to determine IC50

250 cells were plated in 96-well plates (BD Biosciences) and drugs were added on day 1 using a Digital Dispenser (Tecan Männedorf). On day 8, cells were fixed for 10 minutes in 99% methanol and stained with 0,1% crystal violet. After 4 hours, staining solution was removed and plates were washed 4 times with water after which plates were air dried. After air drying plates were scanned after which they were processed for further quantification. For this, 50ul of 10% acetic acid was added for 15 minutes, followed by 150ul of H2O on a shaker. Absorbance was measured using an Epoch Microplate Spectrophotometer (Biotek) and Gen5 software and relative cell survival plots were generated. IC50s were calculated with Prism 8 (GraphPad).

## Results

### Generation of 27 isogenic aneuploid clones

To study the effects of aneuploidy on chromosome instability, we generated a large panel of aneuploid clones derived from an hTERT-immortalized retinal pigment epithelial cell line (RPE-1). As p53 has an important role in aneuploidy tolerance [36, 45, 46] and is mutated in

the majority of cancers, aneuploid clones were generated under a stable shRNA knockdown of p53 [36]. For this, a total of ten 384-well plates (3840 wells) in 5 independent experiments were seeded with single cells that were pretreated overnight with a low dose of MPS1 and CENP-E inhibitors to induce chromosome missegregations. Each well was subsequently analyzed and approximately 650 wells contained a single cell by visual inspection. About ~6% of these cells made it to a full clone and only a subset of those had a confirmed aneuploid karyotype. With this approach, we obtained a total of 27 clones containing one or more *de novo* aneuploidies (since the gain of chromosome 10q and the gain of chromosome 12 are common events in parental cells, these were excluded as *de novo* events; for more details see Materials and methods). A subset of clones contained solely whole chromosome imbalances (S1 Fig), whereas others contained segmental abnormalities, sometimes in combination with whole chromosome aneuploidies (S2 Fig). Segmental aneuploidies have undergone DNA breaks and potentially aberrant DNA repair resulting in abnormal chromosomes. As this could complicate the analysis of the sole effect of aneuploidy on CIN, we first focused on clones exclusively harboring whole chromosome aneuploidies with clean CNV profiles (15 clones, S1 Fig), as these clones are less likely to have experienced chromosome damage during their generation. As a control, we selected two single cell derived clones (C1 and C2) that displayed a karyotype comparable to the parental cell line but underwent the same procedure as the other established aneuploid clones.

## Aneuploid clones show a spectrum of different doubling times and missegregation rates

We set out to characterize our various aneuploid clones. One of the known consequences of aneuploidy is reduced cellular fitness and proliferation [47–49]. To assess the proliferation rates of our clones, we determined the doubling times with live-cell imaging. As expected, the majority of our aneuploid clones displayed impaired growth compared to the euploid controls (Fig 1A). The proliferation impairment was heterogeneous between the different clones with doubling times ranging from 16h to 72h. As we aim to shed light on the relationship between whole chromosome aneuploidy and CIN, we next evaluated the levels of CIN in our clones by live cell imaging (Fig 1B). The parental cell line and the two control clones displayed a basal level of segregation errors of ~7–10%. All levels of missegregations exceeding this basal level of 10% were define as enhanced CIN. When assessing the levels of CIN in our clones using live-cell imaging, we found that aneuploid clones showed a spectrum of missegregation rates, ranging from ~5–70%, involving different types of mitotic errors (Fig 1A, S1–S3 Movies). Most prominently, we observed an increase in chromatin bridges amongst the majority of aneuploid clones. Moreover, a subset of clones displayed increased lagging DNA in anaphase. As live cell imaging did not allow us to distinguish between lagging chromosomes containing a centromere or acentric fragments, we performed fixed analysis of a subset of clones showing an increased level of the lagging DNA category. Fixed analysis confirmed the CIN phenotype that we observed using live-cell imaging in these clones, with the largest increase observed in chromatin bridges and lagging DNA. Interestingly, in most analyzed clones the majority of lagging DNA consisted of centromere-containing chromosomes and only a small fraction of acentric fragments could be observed (S3A Fig). These observations demonstrate that imbalances of whole chromosomes can trigger a spectrum of CIN levels, comprising different types of segregation errors. However, it is important to note that a subset of clones harboring whole chromosome aneuploidies did not show any increase in segregation errors.

To understand if CIN could be an underlying cause for decreased cellular fitness we evaluated the correlation between doubling times and missegregation rates. Interestingly, we found

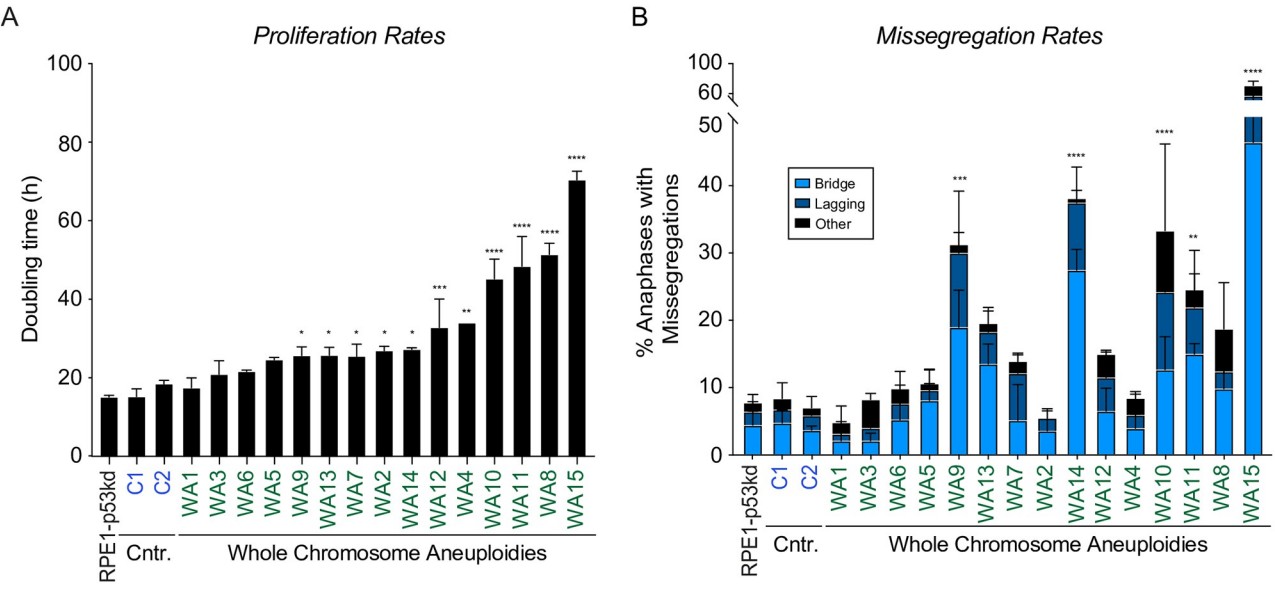

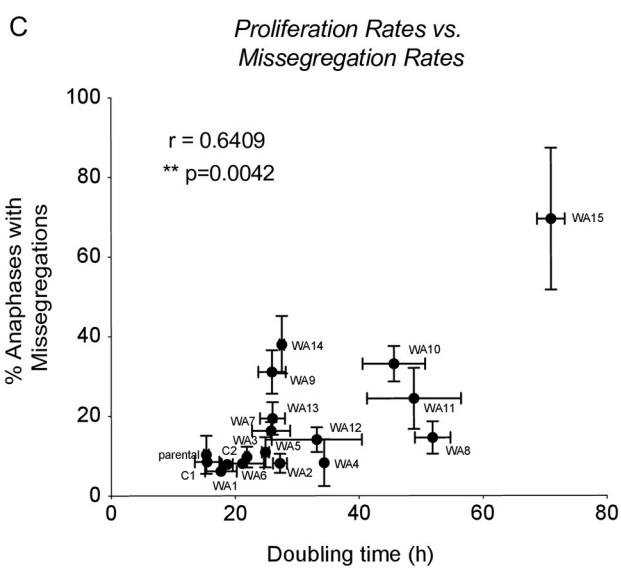

**Fig 1. Aneuploid clones show a spectrum of different doubling times and mis-segregation rates.** A. Doubling time of the parental cell line (labeled in black), two euploid clones (blue) and aneuploid clones (green) determined via live cell imaging. Error bars indicate standard deviation. An ordinary one-way ANOVA was performed between clones and parental cells. P-values are assigned according to GraphPad standard. B. Chromosome missegregation rates determined by live cell imaging of parental RPE-1 p53KD cells (labeled in black), euploid (blue) and aneuploid (green) clones from S1 Fig, divided into three subcategories: lagging DNA, anaphase bridges and others (multipolar spindle, polar chromosome, cytokinesis failure, binucleated cell). All conditions were analyzed blinded. Bars are averages of at least 2 experiments and a minimum of 50 cells were filmed per clone. Error bars indicate standard deviation. An ordinary one-way ANOVA was performed between parental and clones. P-values are assigned according to GraphPad standard. C. Spearman correlation between the proliferation rate as measured in A and the missegregation rates as a percentage of anaphases as measured in B.

a significant positive correlation (Fig 1C, p = 0.0042). However, important to note is that some of our chromosomally stable clones display a clear growth impairment (e.g. clone WA2 and WA4, compare Fig 1A and 1B), while other highly unstable clones seem to have a milder proliferation defect (e.g. clone WA9 and clone WA14). This suggests that although CIN and proliferation impairment are both consequences of aneuploidy that correlate to each other, it is unclear if they have a direct causative relationship.

## Proliferation defects are linked to gene imbalances

Previously, it has been suggested that the impaired proliferation rates in aneuploid yeast cells are largely determined by the increased dosage of coding genes, as additional non-coding DNA did not cause a growth defect [47]. Importantly, a more severe growth impairment could be observed when larger or more chromosomes were gained [47]. The effect of chromosome losses has not been addressed as the studied yeast model contains a haploid genome to start with. We set out to determine if the degree of aneuploidy could also explain the variation in proliferation rates observed in our mammalian cell system. We calculated the degree of aneuploidy by determining the total number of coding genes that are imbalanced per clone. Indeed, we found a positive correlation between doubling times and number of imbalanced genes (Fig 2A, left panel, p = 0.0005). When separating the imbalanced genes in lost and gained genes we found that proliferation rates were more affected by the gain than by the loss of genetic material, as only the gained genes showed a significant correlation to proliferation rates (Fig 2A, middle and right panel, p = 0,005 and p = ns). However, neither the gained genes nor lost genes provided additional information about the growth rate over the number of imbalanced genes, as the number of imbalanced genes could fully explain the two other correlations (conditional Spearman tests p = 0.37 (gained genes vs. growth, covariate imbalanced genes) and p = 0.95 (lost genes vs. growth, covariate imbalanced genes, respectively). Therefore, our data suggests that the reduced proliferation rates are most likely a consequence of gene imbalances.

## CIN rates are explained by gained genes rather than by imbalances

Next, we investigated the relationship between the degree of aneuploidy and the levels of CIN. Studies in yeast suggested that instability is also a consequence of dosage changes in coding genes as the addition of non-coding DNA does not instigate a CIN phenotype while extra coding DNA does [24]. It was previously shown that the more cells deviate from their true euploid state, the more unstable they become [25, 26]. However, a clear correlation between the number of gained material and CIN has not been found [24]. Here, we found a significant correlation between the number of imbalanced coding genes and missegregations rates in our mammalian cell system (Fig 2B, left panel, p = 0.0004). Interestingly, and unlike the correlation between doubling times and degree of aneuploidy, this correlation further improved when only tested against the number of gained coding genes (Fig 2B, middle panel, p<0.0001). There was no significant correlation between the number of lost coding genes and CIN levels (Fig 2B, right panel). Importantly, the correlation between imbalanced genes and CIN rates could be fully explained by the number of gained genes (conditional Spearman test for imbalanced genes after correction for gained genes is p = 0.50), but the correlation of gained genes with CIN rates remained significant after correcting for the total number of imbalanced genes (conditional Spearman test p = 0.001). Together, these data suggest that gaining extra coding genetic material can lead to CIN while losing coding genetic material does not significantly contribute to this phenotype. Moreover, it suggests that proliferation rates and CIN rates are two independent features of aneuploidy that have different underlying causes.

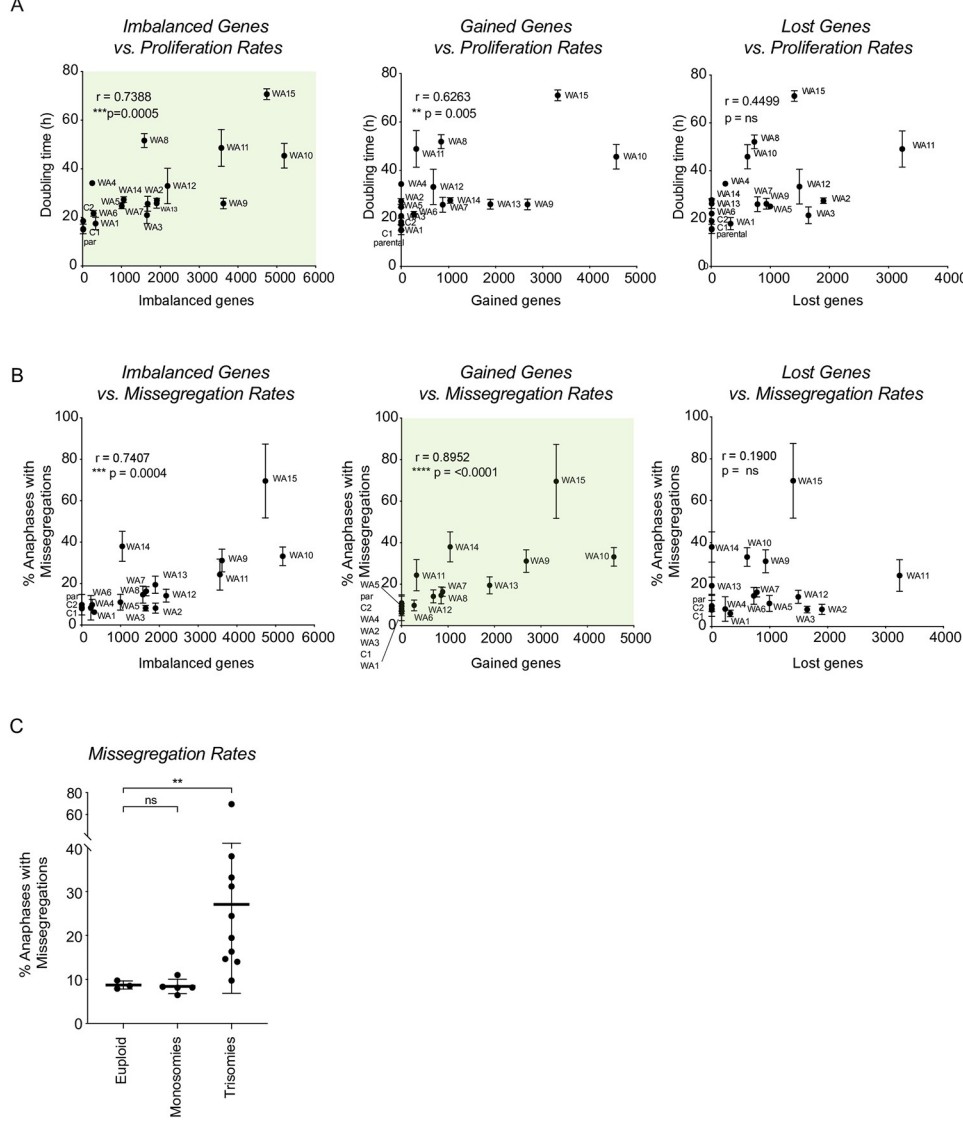

**Fig 2. Both monosomies and trisomies decrease cellular fitness.** A. Spearman correlation between the total number of imbalanced, gained and lost coding genes per clone and proliferation rates as measured in Fig 1A. Error bars indicate standard deviation. Light green panels indicate the preferred model. B. Spearman correlation between the number of imbalanced, gained and lost coding genes and the level of CIN as percentage of total number of anaphases as determined in Fig 1B. Error bars indicate standard deviation. Light green panels indicate the preferred model. C. Missegregation rates described in A, classified in three different categories: euploid clones, clones harboring only monosomies and clones harboring trisomies (sometimes with monosomies in the background). Lines show the mean; error bars indicate standard deviation. The Mann-Whitney U statistical test was used to compare differences in CIN between groups.

In line with the absence of a correlation between the loss of genetic material and CIN rates, we found that clones that did not induce instability mostly involved clones with only monosomies, whereas clones with increased instability exclusively involved clones harboring trisomies. Indeed, when classifying clones by their type of karyotype aberration, the monosomic clones (referred to as monosomies) did not display increased chromosomal instability compared to the control cell lines, while clones that harbor at least one trisomy (referred to as trisomies,

including simple trisomies and more complex karyotypes) in most cases induced CIN to different extents (Fig 2C). This was not due to increased karyotype complexity, as when comparing simple trisomies (harboring one or two gains with no additional aneuploidies) with simple monosomies (harboring one or two losses with no additional aneuploidies) we still observed major differences between trisomies and monosomies. Most strikingly, trisomy clone WA14 has a larger number of gained genes as compared to the number of lost genes in WA2 and WA3 but WA14 displays extremely high CIN levels, whereas WA2 and WA3 did not show CIN (S3B Fig). This further underlines our hypothesis that extra genetic material drives CIN while we did not find evidence that loss of genetic material does so.

## Interfering with proteostasis induces CIN

To further investigate the relation between CIN and gained coding genes, we aimed to elucidate a potential causal relationship. It has been extensively shown that gaining extra coding DNA leads to an excess of proteins being expressed from the involved chromosome [9, 32, 50–52]. Both chromosome gains and losses lead to specific protein imbalances. However, chromosome gains result in the enhanced expression of many genes simultaneously and can cause a general overburden of the protein folding and degradation pathways in an attempt to maintain proteostasis. We therefore hypothesized that a general stress response associated with protein overexpression could be a direct or indirect underlying cause for the CIN phenotype in trisomic clones and would explain the absence of CIN in monosomies. First, we aimed to investigate if inducing proteotoxic stress would be sufficient to drive CIN. To achieve this, we interfered with proteostasis in parental cells using the chaperone protein Hsp90 inhibitor 17-AAG or the proteasome inhibitor MG132, using low doses that did not prevent cells from entering mitosis (S4A Fig). We found that cells treated with either one of the inhibitors showed missegregation rates comparable to the ranges we observed in our clones (Fig 3A). These data suggest that interfering with protein folding or protein degradation in parental cells is sufficient to trigger a CIN phenotype.

As chemically interfering with proteostasis is sufficient to drive CIN, we were wondering if our trisomic clones indeed experience proteotoxic stress that could explain their CIN. To investigate this, we first evaluated autophagic flux, by examining the conversion between LC3B-I to LC3B-II on autophagosomes [53, 54]. This conversion indicates an activated autophagy pathway which aids in the degradation of misfolded/unfolded proteins upon proteotoxic stress and has previously been shown to be upregulated in trisomic cell lines and in cells where aneuploidy was induced with inhibitors [32, 55]. We tested 3 of our trisomic clones that displayed a variety of CIN levels but we could not detect a significant increase in LC3B-II in these clones (S4C Fig). It needs to be noted that we could also not observe increased LC3B-II levels in parental RPE-1 cells treated with low doses of proteostasis inhibitors that were sufficient to induce CIN (S4B Fig), suggesting that this assay might not be sensitive enough to detect low levels of proteotoxic stress or that our clones in fact do not increase autophagic flux efficiently. As a second effort, we set out to test our cells for sensitivity to the Hsp90 inhibitor 17-AAG. A previous study has shown that trisomic MEFs are more sensitive to this compound [34]. Interestingly, the clone with the most trisomies (clone WA10, trisomic for 5 chromosomes) indeed displayed the increased sensitivity to this compound (S4D Fig, p = 0,0097). However, for other trisomic clones we found no increased sensitivity. In fact, some clones showed a similar sensitivity or were even more resistant compared to parental cells. Also, two of the monosomic clones displayed enhanced sensitivity compared to parental cells (S4D Fig). Importantly, the number of gained genes did not correlate with the sensitivity to the drug (R = -0.3, p = n.s.) and neither did the number of imbalanced genes (R = 0.06, p = n.s.). Taken together,

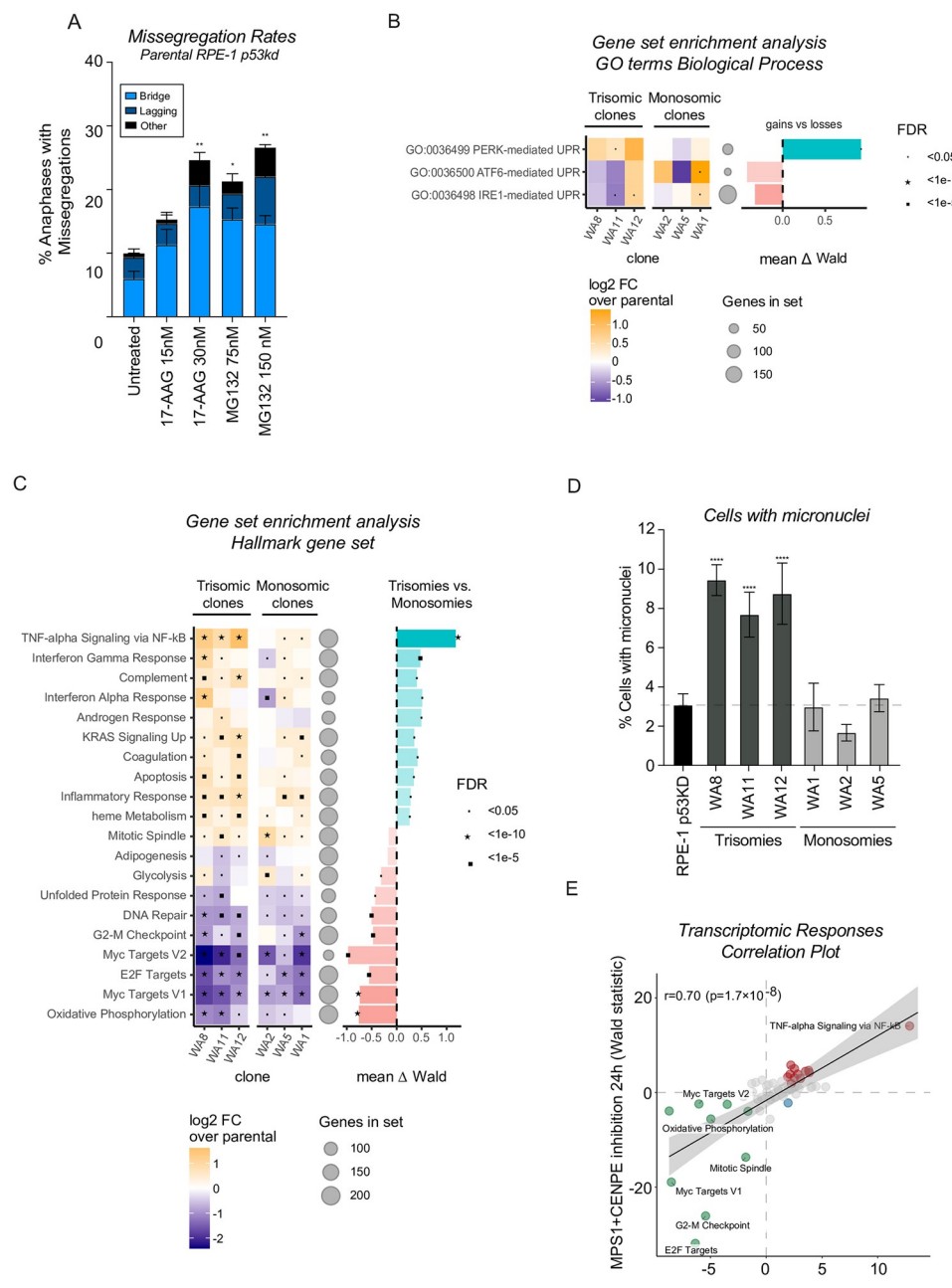

**Fig 3. Different transcriptomic responses to trisomies and monosomies.** A. Chromosome missegregation rates of RPE-1 parental p53KD cells untreated or treated with low doses of the indicated inhibitors for 24 hours determined as in Fig 1B. The experiment was performed in triplo and 50 cells were analyzed per condition per experiment. Error bars indicate standard deviation. B. Gene set enrichment analysis (GSEA) specifically for three different UPR branches, evaluating their up and downregulation in trisomic and monosomic clones compared to parental cells (left graph). Two replicates of every clone were sequenced and the Log2 fold change (FC) was determined compared to parental cells. The false discovery rates (FDR) are indicated with symbols. Differences between trisomies and monosomies were determined by Wald statistical testing. C. Gene set enrichment analysis (GSEA) of RNA sequencing data, evaluating up and downregulated hallmarks in trisomy clones and monosomy clones compared to parental (left graph). Two replicates of every clone were sequenced and the Log2 fold change (FC) was determined compared to parental cells. The false discovery rates (FDR) are indicate with symbols. Largest differences between trisomies and monosomies are shown on the right. Differences in hallmarks between trisomies and monosomies were determined by Wald statistical testing. D. Percentage of cells harboring micronuclei, as determined via snapshots from live-cell imaging data. 2 experiments were analyzed per clone, a minimum of 150 cells was analyzed per clone per experiment. Bars show the

average of 2 experiments; error bars indicate standard deviation. E. Correlation between upregulated and downregulated hallmarks in acute aneuploidy and the upregulated and downregulated hallmarks showing the largest difference between trisomies and monosomies determined by Wald statistical testing. Red dots indicate significantly upregulated, and green dots indicate significantly downregulated hallmarks, blue means not significant.

interfering with proteostasis in parental cells induces similar CIN levels as observed in our trisomic clones. However, two directed experiments could not provide strong evidence that proteotoxic stress (at least activation of the autophagy pathway or sensitivity to protein folding inhibitors) is consistently present in these clones. It is possible that the levels of proteotoxic stress are below detection limit, or that we assessed aspects of proteostasis that are not relevant to the CIN phenotype.

## Trisomic clones show preferential activation of PERK-mediated unfolded protein response

As we could not directly show the presence of proteotoxic stress in our trisomic clones by two directed assays, we decided to perform RNA sequencing as an unbiased approach to find evidence for proteotoxic stress in trisomic cells. Moreover, with this approach we can identify the most prominent differences between monosomic and trisomic clones that could be attributed to the induced CIN in trisomic clones. For this we selected 3 monosomic clones (WA1, WA2 and WA5) and 3 trisomic clones (WA8, WA11 and WA12). We analyzed differential expression compared to parental cells for each clone and determined the most significant genes and gene sets between the two subcategories (trisomies versus monosomies). The most relevant Hallmark for proteotoxic stress is the unfolded protein response (UPR), which encompasses a transcriptional and translational response to endoplasmic reticulum (ER) stress resulting in repressed translation and apoptosis depending on the extent and duration of the response. Unexpectedly, we found that the majority of aneuploid clones in fact displayed a downregulated UPR Hallmark compared to the parental cells (Fig 3B). As the UPR consists of three main branches; PERK, ATF6 and IRE1 pathways, we decided to evaluate each pathway separately. This analysis showed that the PERK branch of the UPR was significantly upregulated specifically in our trisomic clones, whilst the other branches showed inconsistent results between trisomies and monosomies (Fig 3B). Furthermore, we found two phosphatases that are part of a central negative feedback loop in PERK signaling, namely PPP1R15A (also known as GADD34) and PPPR15B (also known as CreP), that are important for the dephosphorylation of eIF2, to be upregulated significantly in our trisomic clones (p = 0.027 and p = 0.001). This indicates that ER-stress might indeed be activated in trisomic clones and not in monosomic clones and this specifically involves the PERK/ATF4-branch. Taken together, we find 1) a strong correlation between gained genes and CIN, 2) that interfering with proteostasis is sufficient to induce CIN and 3) that the PERK pathway is specifically upregulated in trisomic clones. Potentially, this indicates that ER-stress can be linked to CIN.

## Trisomic clones show a differential expression profile related to CIN

Besides focusing on the differential regulation of ER stress, we decided to analyze our data to find the pathways that were affected most prominently in trisomic clones as compared to monosomic clones to potentially reveal additional links between chromosome gains and CIN. We found that several cell cycle-related hallmarks (E2F targets, Myc targets, G2-M checkpoint) were downregulated in all our aneuploid clones but more prominently in the trisomic clones (Fig 3C), which agrees with the decreased proliferation rates found in these clones.

Interestingly, we found that the trisomic clones showed upregulation of several hallmarks associated with an inflammatory response, such as TNF-alpha signaling via NF-kB, interferon responses, complement and inflammatory response. These responses were either absent, downregulated or only mildly activated in the monosomic clones (Fig 3C). The upregulation of specific cytokines was confirmed by qRT-PCR in our trisomic clones (S5A Fig). Rather than a cause for CIN, the pronounced upregulation of inflammatory responses could be a consequence of the elevated levels of CIN. It is known that CIN can induce such a response via the cGAS-STING pathway, as a result of micronuclei rupture or chromatin bridges [56, 57] or via cGAS-independent activation of NF-kB [58]. Indeed, and in line with our missegregation data, we observed an increase in micronuclei in clones harboring trisomies but not in our monosomic clones (Fig 3D). These data suggest that the elevated CIN could be responsible for driving parts of the transcriptional responses in trisomic clones. To understand which transcriptomic responses in trisomies are explained by the enhanced CIN levels, we performed RNA sequencing of parental cells treated with a low dose of Mps1 and CENP-E inhibitors for 24 hours to induce acute chromosomal instability. We next investigated which hallmarks that were preferentially affected in trisomic clones correlated to the hallmarks affected in parental cells with acute CIN, to determine the responses that are likely driven by CIN. Indeed, we found an overall significant correlation between the hallmarks that were preferentially affected in trisomic clones compared to parental cells with acute CIN (Fig 3E). Most prominently, we found a very strong upregulation for TNF-alpha signaling via NF-kB, which was confirmed by checking upregulation of specific cytokines via qRT-PCR (S5B Fig). Thus, the inflammatory response that is unique to trisomies is indeed likely explained by the fact that these clones are more CIN.

As a more direct proof for CIN driving this inflammatory response in our clones, we aimed to reduce CIN in various clones. To this end, we treated our clones with UMK57, an MCAK agonist. It was shown that this compound reduces lagging chromosomes in cancer cell lines due to increasing microtubule dynamics and therefore allows for the resolution of stable erroneous attachments [59]. In line with this study, we showed that a working concentration of 100 nM indeed reduced the amount of lagging DNA in anaphase in two CIN cell lines: U2OS and SW620, while only minimally affecting cell viability (S6A and S6C Fig). However, using the same sublethal concentration in our clones, we did not observe a consistent reduction in lagging DNA (S6A and S6E Fig). This suggests that the lagging chromosomes in our clones are unlikely caused by hyperstable kinetochore-microtubule attachments. Most importantly, UMK57 was unable to rescue chromatin bridges in both our aneuploid clones and the two CIN cell lines (S6B and S6D Fig). Since chromatin bridges can also result in micronuclei, and can activate cGAS/STING, we concluded that treatment with UMK57 is not suitable to test the direct relationship between CIN and the observed inflammatory response.

In order to get a more fine-grained understanding of the transcriptional changes between trisomies and monosomies, we also quantified which Gene Ontology (GO) Biological Process categories were expressed at different levels. We found that pathways associated with translation and ribosome biogenesis were downregulated in all clones (rRNA processing, translation initiation) (S5C Fig). Although in monosomies this downregulation was previously attributed to haploinsufficiency of ribosomal genes [60], our data suggest that chromosome gains can possibly also lead to deregulation of these pathways, possibly due to a general aneuploidy-induced stress response [61]. Taken together, trisomies and monosomies have some general aneuploidy induced transcriptional responses, but some differentially expressed Hallmarks are likely a consequence of CIN.

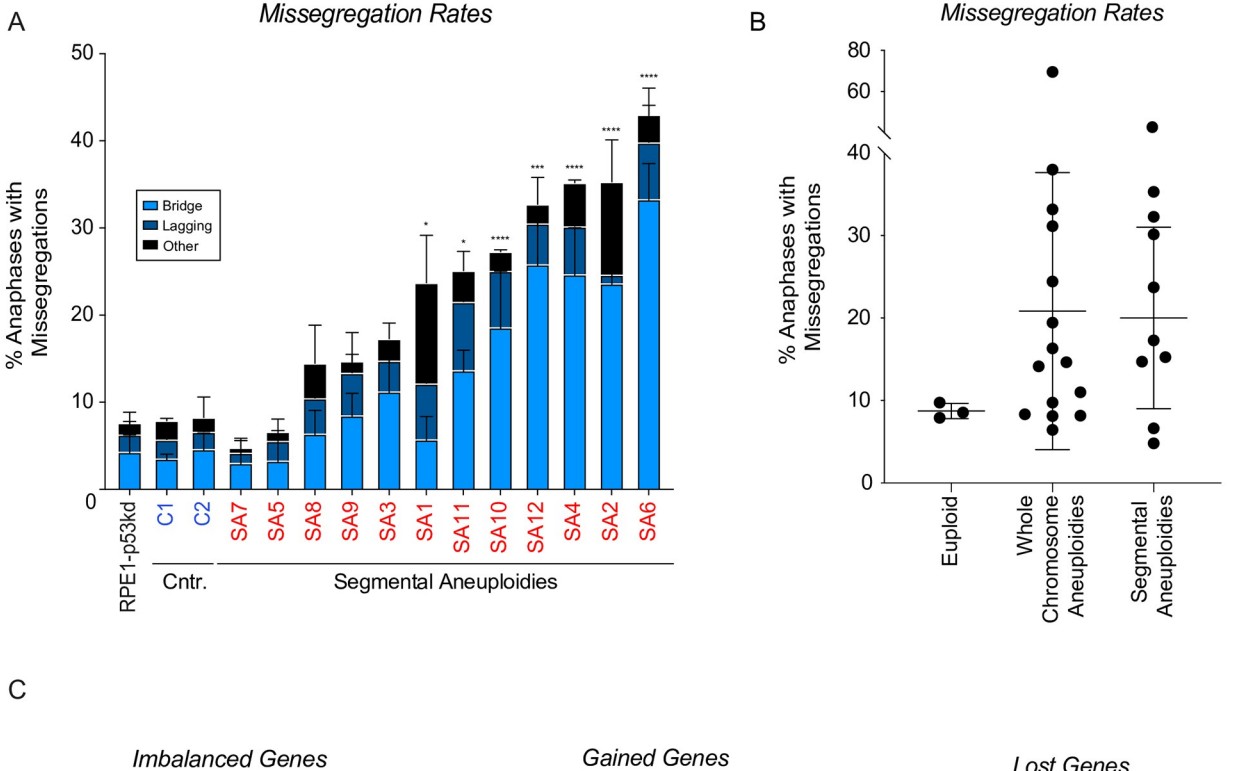

**Fig 4. Segmental chromosome aneuploidies can induce CIN.** A. Chromosome missegregation rates determined by live cell imaging of parental RPE-1 p53KD cells (labeled in black), euploid clones (blue) from S1 Fig and the segmental aneuploid clones (red) from S2 Fig, categorized in three different categories: lagging chromosomes or chromosome fragments, anaphase bridges and others (multipolar spindle, polar chromosome, cytokinesis failure, binucleated cell). All conditions were analyzed blinded. Bars are averages of at least 2 experiments and a minimum of 50 cells were filmed per clones. Error bars indicate standard deviation. An ordinary one-way ANOVA was performed between parental and clones. P-values are assigned according to GraphPad standard. B. Missegregation rates as determined in 1A and 4A, classified in euploid clones, clones harboring whole chromosome aneuploidies, and clones harboring segmental aneuploidies. Lines show the mean; error bars indicate standard deviation. C. Spearman correlation between the number of imbalanced, gained and lost coding genes and the level of CIN as determined in A. Error bars indicate standard deviation.

## Segmental chromosome aneuploidies can induce CIN via BFB-cycles

Besides clones with whole chromosome aneuploidies, we also obtained clones carrying segmental aneuploidies with or without additional whole chromosome imbalances (12 clones, S2 Fig). As mentioned above, we decide to separate these clones from clones harboring only whole chromosome aneuploidies as these clones have experienced DNA damage due to

chromosome breakage at the time of their generation. When analyzing CIN levels in these clones, we again observed that not all segmental aneuploidies instigate a CIN phenotype (e.g. clones SA5 and SA7) (Fig 4A). Consistently with our clones harboring whole chromosome monosomies, clone SA5 and SA7 harbor solely segmental losses (S2 Fig). Moreover, we could observe that a spectrum of instabilities was induced by the clones harboring segmental aneuploidies (Fig 4A and 4B). Consistently, we observed a strong correlation between the number of gained genes and CIN rates (Fig 4C, middle panel, p = 0.0015). Again, there was no significant correlation between lost genes and CIN levels, as observed in clones harboring whole chromosome aneuploidies (Fig 4C, right panel).

One of the segmental clones (SA6) with high CIN rates displayed aberrant sequence reads on the q-arm of chromosome 3 as observed by CNV sequencing (S2 Fig). We selected this clone for further characterization. We performed single-cell sequencing to determine the copy number variations per cell to get more insight into the variation between cells (Fig 5A and 5B). Besides the expected gain of 10q, only few instabilities could be observed in the parental cells, in line with the live cell imaging data (Figs 5A and 1A). Clone SA6 displayed the imbalances observed by CNV, namely a partial loss of chromosome 3 and 20, a loss of chromosome 13 and a gain of chromosome 14 and 19 (Fig 5B). Strikingly, the extent of the loss of the terminal part of chromosome 3 was different in every single cell analyzed, a pattern also observed for the partial loss of chromosome 20 in a subset of the cells. All these observations were confirmed and further detailed by performing COBRA-FISH analysis that allows for the visualization of all different chromosomes (Fig 5C and 5D). 95% of the cells displayed a derivative chromosome resulting from a translocation between 10 and 14. Also, we found in 15/40 (37,5%) analyzed metaphase cells that chromosome 3 was affected by translocations, dicentric chromosome formation or telomeric associations. The most frequent abnormality observed related to chromosome 3, also involved chromosome 20, which often formed a derivative dicentric chromosome at various breakpoints (Fig 5A and 5D). Probing for specific locations of chromosome 3 showed that one copy of chromosome 3 had lost both signals representing the q-arm in clone SA6 (Fig 5E). The high frequency of dicentric chromosome formation along with ongoing loss of parts of chromosome 3 and 20 is consistent with an ongoing breakage-fusion-bridge (BFB)-cycle [62]. These findings might suggest that an initial missegregation led to the breakage of chromosome 3 and 20, and consequent fusion of the broken ends resulted in the formation of a dicentric chromosome, thereby triggering ongoing BFB-cycles. These data show that besides the general effects of genomic imbalances, segmental imbalances can instigate the formation of dicentric chromosomes as a result of fusion of the broken chromatin fragment with another chromosome and the consequent breakage-fusion-break cycles, which lead to CIN. We do not think that BFB-cycle are exclusive to clones with segmental aneuploidies. However, the fact that the generation of a dicentric chromosome was likely the initiating event in this segmental aneuploid clone, allowed for the easy identification of the BFB-cycle and the specific chromosomes involved, as this was present in the majority of the population. Clones with whole chromosome aneuploid aneuploidies also display a high number of bridges which might reflect BFB-cycles but can also be a starting point for BFB-cycles when these bridges break or fragment and relegate to form dicentric chromosomes. However, in clones with whole chromosome aneuploidies these are random events and are likely to be selected out in the population and are therefore more difficult to be identified.

## Discussion

Aneuploidy and CIN are both prevalent features of tumors that highly correlate with each other [63]. It has been proposed that aneuploidy can be a driver of CIN. Indeed, it has been

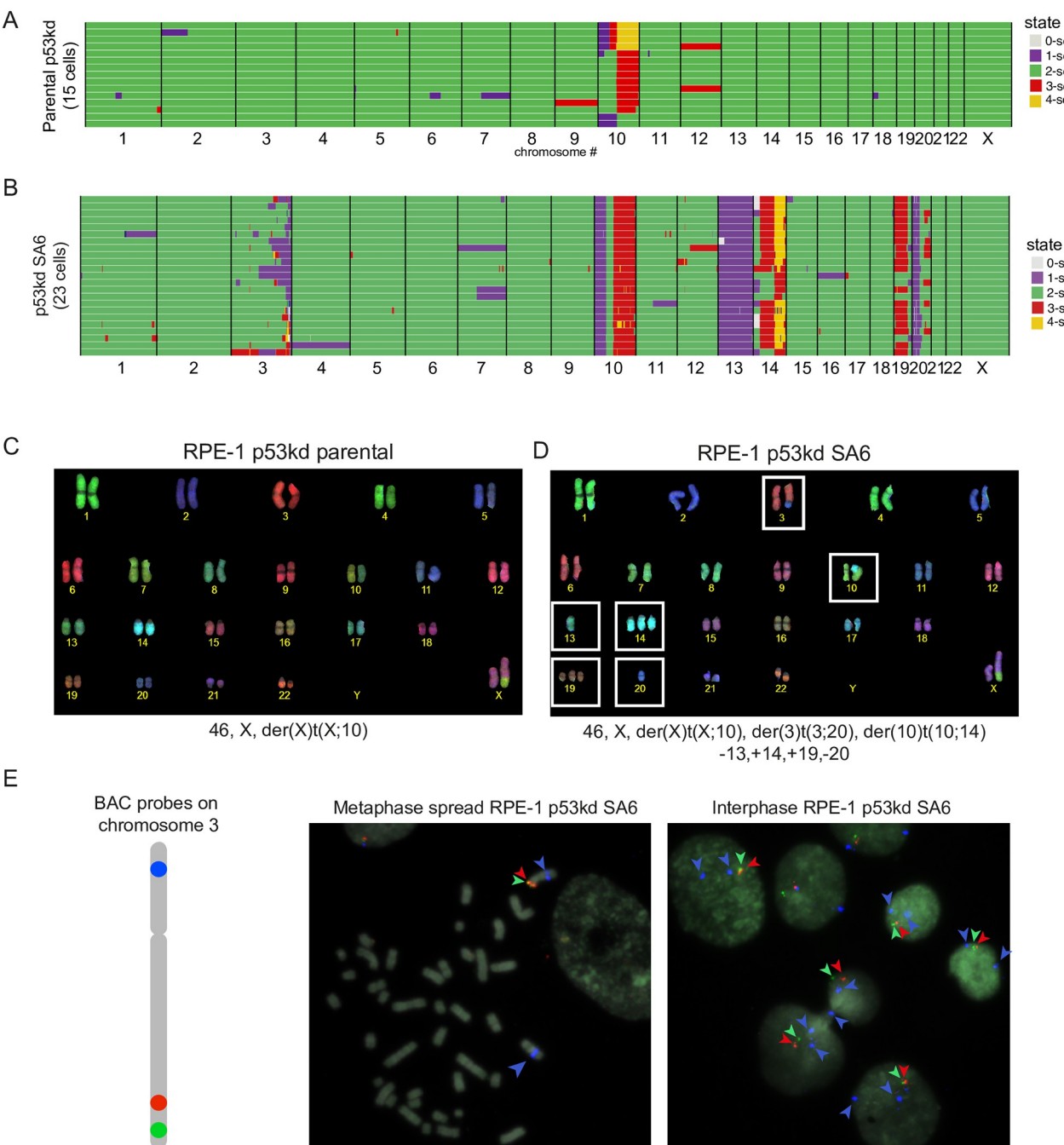

**Fig 5. Segmental aneuploidies can lead to the onset of BFB-cycles via dicentric chromosome formation.** A. Genome-wide chromosome copy number profiles of parental RPE-1 p53kd as determined by single-cell sequencing. Green indicates disomic chromosome regions, purple monosomic, red trisomic and yellow tetrasomic. Gain of the q-arm of chromosome 10 and occasional gain of chromosome 12 is expected. B. Genome-wide chromosome copy number profiles of clone SA6 as determined by single-cell sequencing as in A. Different patterns of aneuploidy are seen in chromosome 3. C-D. Representative images of chromosome spreads labeled with combined binary ratio labelling–fluorescence in situ hybridization of parental RPE-1 cells and clone SA6. The corresponding karyotypes are indicated below the images. White boxes indicate both numerical and segmental abnormalities specific to SA6. E) Metaphase (left) and interphase (right) representative images of cells from clone SA6, using BAC probes specific for different locations throughout the long arm (in red and green) and the short arm (in blue).

shown that extra chromosomes can instigate a CIN phenotype (Sheltzer et al., 2011; Zhu et al., 2012; Duesberg et al., 1998; Nicholson et al., 2015; Passerini et al., 2016), however there are also examples where aneuploidy does not induce CIN [25, 35, 64]. Moreover, there are cases where aneuploidy and CIN do not co-occur in tumors [63, 65]. This indicates that there must be specific aspects of aneuploidy that can contribute to CIN. To understand this, we set out to systematically investigate the impact of many different aneuploid karyotypes on CIN. While studies focusing on the consequences of chromosomal gains have been done extensively, research investigating the cellular consequences of monosomies or segmental aneuploidies have been limited to date. By creating aneuploid clones harboring many unique karyotypes, we were able to extensively evaluate the link between aneuploidy and CIN.

## Gene imbalances lead to impaired growth

In line with previous studies, we found that the cellular fitness of the vast majority of our aneuploid clones was decreased [24, 32, 47–49]. We found a significant correlation between the number of imbalanced genes and reduced proliferation rates, and both the gains and losses of genes contributed to this correlation. In line with this, we found that both monosomic and trisomic clones were growth perturbed. We conclude that CIN is not a direct cause for slow growth as some fast-growing clones are highly unstable and some of the monosomic clones displayed a growth impairment although they are not CIN. This is in line with previous studies in yeast that found no correlation between CIN levels and growth impairment [24, 25]. It needs to be noted that the induction of acute CIN did cause a downregulation of cell cycle associated hallmarks (Fig 3D), which might indicate that CIN has a minor contribution to the slow growth phenotype. However, our findings indicate that the reduced growth of aneuploid cells is likely driven by the dosage changes of specific genes caused by both gains and losses and is not directly caused by the CIN phenotype.

## Monosomies are chromosomally stable

When evaluating clones harboring whole chromosome aneuploidies, we showed that aneuploidy *per se* is not always sufficient to induce CIN, as a subset of our clones were chromosomally stable. Remarkably, we found that clones that only harbor monosomies do not instigate a CIN phenotype while most clones harboring trisomies do. This observation might seem surprising at first as there are many studies that showed that aneuploidy leads to CIN. However, the specific effects of monosomies versus trisomies on CIN have not been extensively explored before as most model systems studying the effects of aneuploidy solely involve chromosome gains [24, 25, 28, 30]. Also, in contrast to our findings, it was recently documented that certain monosomies can in fact trigger a CIN phenotype [60]. There might be several explanations for this apparent discrepancy. First, it was reported that there are additional aneuploidies present in the background of a subset of the monosomy clones. This makes the direct comparison between clones difficult. For example, there are two clones harboring a chromosome 13 loss, however a segmental gain of chromosome 22 is present in the background of the p53KO clone that does display CIN, while no background aneuploidies are observed in the p53KD clone, that does not display CIN. Such background aneuploidies could contribute to the observed instability as chromosome gains can instigate a CIN phenotype. Moreover, in the study by Chunduri et al, they observed chromatin bridges only in p53KO cells and not in monosomies generated in a p53kd background, that we used to generate our clones. Therefore, we cannot rule out that there is a role for minimal levels of p53 that are potentially left in our monosomic clones that would protect against CIN. However, this is unlikely as we do observe high CIN levels in the trisomic clones that were generated in the same background.

Importantly, the fact that monosomies are chromosomally stable cannot simply be explained by a difference in karyotype complexity as trisomic clones with karyotype complexities similar to the monosomic clones do show increased CIN (for example compare WA3 to WA8). Finally, since we only investigated a limited number of monosomic clones, it is possible that losing certain chromosomes can indeed lead to instability for instance due to certain haplo-insufficient genes involved in chromosome segregation that are present on these lost chromosomes.

## Trisomies cause CIN levels that correlate with the number of gained genes

Interestingly, we found a strong correlation between the levels of CIN and the number of gained coding genes. It has been shown before in aneuploid yeast strains that the negative consequences of aneuploidy, including genomic instability, are due to the presence of extra genes and not the presence of extra DNA [24, 47], suggesting that CIN is indeed induced by altered expression of genes and not simply a consequence of having extra chromosomes that need to participate in mitosis. However, these studies did not find a correlation between the number of gained material and instability, suggesting that the underlying mechanisms driving CIN might differ between yeast and mammals.

So why do chromosome gains result in CIN and monosomies do not? Both gaining extra genetic material as well as losing genetic material will result in imbalances of proteins. Such expression changes will affect complex stoichiometry in specific cases. Cells deal with the excess of unincorporated proteins by performing dosage compensation [47, 51, 66, 67]. This has been shown to occur both in trisomies [32] as well as in monosomies [60]. Although both monosomies and trisomies perform dosage compensation, the critical difference between monosomies and trisomies might be their distinct mechanism for dosage compensation and the associated stress pathways that are induced. Imbalances caused by monosomies could be resolved by upregulating the respective haploid gene products by decreasing protein turnover and/or increasing protein production while trisomies rather increase protein turnover by enhancing their degradation, often coinciding with enhanced levels of unfolded protein, ER stress and aggregate formation [5, 50, 51, 66, 68]. In line with this, a recent study by Schukken et al., investigated the extent of dosage compensation in human cancer cell lines both on RNA level as well as protein level. Their data showed that dosage compensation occurs both upon chromosome gains and losses to a similar extent [69]. Furthermore, they indeed provide evidence that chromosome gains and losses have distinct mechanisms of dosage compensation. We therefore hypothesize that the distinct mechanisms of dosage compensation between chromosomal gains and losses could possibly underlie the differences that we observe in chromosomal instability.

To test if dosage compensation occurs in our clones, we evaluated the correlation between DNA copy number and global RNA expression levels in our clones (S7 Fig). The gene expression levels of most chromosomes scaled with DNA copy number, as most aneuploid chromosomes locate close to the grey dashed line, representing no compensation. We found that the loss of 1 copy of the X-chromosome does not have any impact on the transcriptome in clone WA2. This could either mean full dosage compensation, but it is most likely that this clone lost the silenced copy of the X-chromosome. For a subset of other aneuploid chromosomes, we observed some evidence for dosage compensation on certain chromosomes, both on monosomies (for example chromosome 13 in clone WA1 and WA2; and chromosome 14 in WA12) as on trisomies (chromosomes 12 and 16 in WA8). However, we need to note that the amount of data is limited and that there is also noise observed around the chromosomes that are euploid. Also, we lack the resolution to conclude if there is dosage compensation on the level of

individual, possibly critical, genes. Finally, and most importantly, to make solid conclusions on dosage compensation, we require proteome data as we assume the biggest consequences of dosage compensation come from mechanisms associated with protein degradation.

Degrading proteins is very energy demanding which can have a bigger impact on cells than inhibiting/preventing degradation, which would be sufficient to overcome critical dosage alterations induced by monosomies. In line with this, studies in yeast indicate that the toxicity of individually overexpressed dosage-sensitive genes can be attributed to the enhanced burden on the protein turnover machinery [70, 71]. Indeed, interfering with protein folding and turnover in parental cells using low doses of inhibitors resulted in CIN, suggesting that interfering with the proteostasis machinery might be sufficient to drive chromosomal instability. Moreover, transcriptome analysis showed an activation of the PERK-branch of the UPR and upregulation of its negative regulators specifically in trisomies, suggesting ER stress to be selective for trisomic clones. Interestingly, a recent study found a positive correlation between the activation of the UPR (specifically the PERK-branch) and the degree of aneuploidy in a pan-cancer analysis, further supporting the validity of our observations [72]. However, how ER stress or saturation of the protein turnover machinery could drive CIN remains enigmatic. Future studies should shed light on the mechanisms of dosage compensation in trisomies and monosomies, if this indeed has a different impact on the stress pathways that are activated.

An important remaining question is how can enhanced ER-stress or proteotoxic stress translate into CIN? We can speculate based on the types of CIN we observed in our trisomic clones. The main category of CIN induced in the trisomic clones was chromatin bridges. Bridges can have several causes: replication intermediates, repair intermediates, unresolved sister chromatid catenanes or dicentric chromosome formation (for example caused by telomere fusions). As replication stress has been widely described as a consequence of aneuploidy [73], we speculate that enhanced levels of replication stress could explain the enhanced levels of chromatin bridges in our clones. Furthermore, we found an increase in lagging DNA, that consisted mainly of lagging centric chromosomes in a subset of clones, which could be a result of mitotic errors such as merotelic attachments. Interestingly, such errors were not due to hyper-stable attachments, as destabilizing MT-KT attachments by UMK57 could not rescue these errors. Intriguingly, it is reported that mild replication stress can also cause merotelic attachments due to premature centriole engagement [74]. Thus, it is possible that ER stress translates into replication stress, although this awaits further investigation.

## Trisomies induce a stronger inflammatory response as compared to monosomies

The most prominent difference on transcriptome level between trisomies and monosomies, was the more significant upregulation of an inflammatory response in trisomies. As a similar response was observed in parental cells with induced acute CIN, this suggests that this response is probably a consequence of the elevated levels of CIN in the trisomic clones. However, since we were not able to rescue CIN in our clones using UMK57, we can only speculate which transcriptomic changes are driven by CIN and which are caused by chromosome imbalances. Future work aiming to understand the underlying mechanisms of CIN in our clones could open up new strategies to reduce CIN and to directly test this relationship. Possibly, proteotoxic stress itself can also contribute to an elevated inflammatory response as it has been shown that an overload of the ER with proteins accumulating inside the organelle can trigger an NF-kB response [75]. More research is needed to reveal which exact aspects of CIN or which other factors are responsible for the elevated inflammatory response that we observe in trisomies and how they relate to each other.

## Segmental aneuploidies can result in BFB-cycles

Clones harboring segmental aneuploidies also display a link between the number of gained genes and instability. However, we found that segmental aneuploidies can have additional defects leading to chromosomal instability. By combining single cell sequencing and COBRA--FISH, we found strong evidence of an ongoing BFB-cycle, in at least one clone harboring segmental aneuploidies. This ongoing BFB-cycle was driven by a fusion between chromosome 3 and chromosome 20, resulting in a dicentric chromosome. Although BFB-cycles have been associated with telomere damage [76], we show here that such segmental rearrangements can also arise after chromosome missegregation events resulting in broken chromosomes. Thus, faulty repair of segmental aneuploidies can result in the formation of abnormal derivative chromosomes, thereby leading to ongoing BFB-cycles, which can be an important mechanism for genomic amplifications seen in cancer [77]. It can be expected that abnormal chromosomes also contribute to bridge formation in trisomic clones, for example due to chromotriptic events or broken chromosomes as a consequence of micronuclei formation or chromatin bridges.

## Concluding remarks

Together, our findings show that aneuploidy *per se* does not induce chromosomal instability. We observed that clones harboring trisomies show various levels of CIN while monosomies are chromosomally stable. These elevated CIN levels correlate with a stronger activation of the inflammatory response in trisomies as compared to monosomies. Interestingly, levels of CIN correlate significantly with the number of gained coding genes. Moreover, inhibiting protein folding or protein turnover pathways in parental cells is sufficient to induce CIN. We hypothesize that excess protein production is putting a burden on the protein turnover machinery and this results in CIN by a yet to be defined mechanism. Finally, we found that segmental aneuploidies can cause ongoing segregation errors by inducing BFB-cycles. This knowledge contributes to our understanding of the relationship between aneuploidy and CIN and how different types of aneuploidy are therefore likely to have different impacts on cancer initiation and development.

## Supporting information

**S1 Fig. Characterization of euploid clones and whole chromosome aneuploid clones.** Genome-wide chromosome copy number profile as determined by CNV-seq of the RPE-1 p53kd parental clone (labeled in black), two euploid clones (labeled in blue) and 10 clones harboring solely whole chromosome imbalances (labeled in green). Chromosome gains and losses were depicted in green boxes. Alterations of chromosome 10 and 12, already present in the parental cells, were not highlighted.
(TIF)

**S2 Fig. Characterization of segmental chromosome aneuploid clones.** Genome-wide chromosome copy number profile as determined by CNV-seq of the RPE-1 p53kd parental clone (labeled in black), two euploid clones (labeled in blue) and 10 clones harboring segmental chromosome imbalances (labeled in red). Chromosome gains and losses were depicted in green boxes. Alterations of chromosome 10 and 12, already present in the parental cells, were not highlighted.
(TIF)

**S3 Fig. Analysis of CIN with simple karyotypes.** Comparison of simple monosomies and simple trisomies. Simple trisomies show elevated CIN while simple monosomies with

comparable levels of gained genes do not. Red dots represent trisomic clones and green dots represent monosomic clones. Missegregation levels as measured in Fig 1B. Error bars indicate standard deviation.

(TIF)

**S4 Fig. No detectable proteotoxic stress in trisomies or in parental cells treated with proteostasis interfering drugs.** A. Doubling time of parental p53KD cells untreated and treated with different concentrations of proteostasis interfering drugs, as measured in Fig 2. Error bars indicate standard deviation. An ordinary one-way ANOVA was performed between parental and clones. P-values are assigned according to GraphPad standard. B. Immunoblot showing conversion from LC3B-I to LC3B-II in parental p53KD cells untreated, treated with 50uM chloroquine and treated with low doses of proteostasis interfering drugs for 24 hours. Loading control is Alpha Tubulin. C. Immunoblot showing conversion from LC3B-I to LC3B-II in parental p53KD cells untreated, treated with 50uM chloroquine to block autophagy as a positive control and 3 different trisomic clones. Loading control is Alpha Tubulin. D. IC50 values of Hsp90 inhibitor 17-AAG as determined by growth assays of parental p53KD cells and various trisomic clones, ordered per number of gained coding genes. Error bars indicate standard deviation. An ordinary one-way ANOVA was performed between parental and clones.

(TIF)

**S5 Fig. Differentially expressed GO Biological processes between Trisomic and Monosomic clones.** A. mRNA levels of inflammatory response cytokines determined via qRT-PCR in trisomic clones. Values were normalized to Actin and are displayed relative to expression levels in parental cells. Bars show mean expression levels; error bars indicate upper and lower limits. Dashed line represents parental expression levels. B. mRNA levels of inflammatory response cytokines determined via qRT-PCR in parental cells treated with different concentrations of Mps1i for 24 hours. Values were normalized to Actin and relative to expression levels in parental cells. Bars show mean expression levels; error bars indicate upper and lower limits. Dashed line represents parental expression levels. C. Gene set enrichment analysis (GSEA) of RNA sequencing data, evaluating up and downregulated GO Biological Processes in trisomy clones and monosomy clones compared to parental (left graph). Two replicates of every clone were sequenced and the Log2 fold change (FC) was determined compared to parental cells. The false discovery rates (FDR) are indicate with symbols. Largest differences between trisomies and monosomies are shown on the right. Differences in hallmarks between trisomies and monosomies were determined by Wald statistical testing.

(TIF)

**S6 Fig. Treatment with UMK57 does not significantly rescue CIN in our clones.** A. Relative cell viability assay in U2OS, SW620 and RPE-1 cells to determine working concentration and compound functionality using Crystal Violet staining. B. Chromosome missegregation rates determined by live cell imaging of U2OS and SW620 cells treated with DMSO or UMK57 (100nM), divided into three subcategories: lagging DNA, anaphase bridges and others (multipolar spindle, polar chromosome, cytokinesis failure, binucleated cell). All conditions were analyzed blinded. Bars are averages of at least 2 experiments and a minimum of 50 cells were filmed per clone. Error bars indicate standard deviation. C. Percentage of only lagging DNA in anaphase in U2OS and SW620 cells. D. Chromosome missegregation rates determined by live cell imaging of RPE-1 parental p53kd cells and 3 different clones treated with DMSO or UMK57 (100nM), divided into three subcategories: lagging DNA, anaphase bridges and others (multipolar spindle, polar chromosome, cytokinesis failure, binucleated cell). All conditions

were analyzed blinded. Bars are averages of at least 2 experiments and a minimum of 50 cells were filmed per clone. Error bars indicate standard deviation. E. Percentage of only lagging DNA in anaphase in U2OS and SW620 cells. Error bars indicate standard deviation. T-test analysis revealed no significant changes between DMSO and UMK57 treatment.
(TIF)

**S7 Fig. Minor dosage compensation when evaluating the correlation between transcriptome and karyotype.** Correlation plotted between gene expression levels and DNA copy number per chromosome. Grey dashed line represents expected value when no compensation is observed. Dotted lines represent 100% compensation or 50% compensation. Upwards pointing arrows indicate trisomy clones, downwards pointing arrows indicate monosomy clones. All chromosomes of each clone are plotted, in representative colors.
(TIF)

**S1 Raw images. Raw image files of western blots.**
(PDF)

**S1 Movie. Example of lagging DNA.**
(AVI)

**S2 Movie. Example of chromatin bridge.**
(AVI)

**S3 Movie. Example of multipolar spindle.**
(AVI)

## Acknowledgments

We thank the Genomics Core Facility of the Netherlands Cancer Institute for sample preparation, data acquisition, and analysis of CNV and RNA sequencing experiments.

## Author Contributions

**Conceptualization:** Dorine C. Hintzen, Mar Soto.

**Data curation:** Dorine C. Hintzen, Mar Soto, Michael Schubert, Bjorn Bakker.

**Formal analysis:** Dorine C. Hintzen, Bjorn Bakker, Karoly Szuhai, Roel J. C. Kluin.

**Funding acquisition:** René H. Medema, Jonne A. Raaijmakers.

**Investigation:** Dorine C. Hintzen, Mar Soto, Michael Schubert.

**Methodology:** Dorine C. Hintzen, Mar Soto, Bjorn Bakker, Diana C. J. Spierings, Karoly Szuhai, Peter M. Lansdorp.

**Project administration:** René H. Medema, Jonne A. Raaijmakers.

**Resources:** Floris Foijer.

**Supervision:** René H. Medema, Jonne A. Raaijmakers.

**Writing – original draft:** Dorine C. Hintzen, Mar Soto.

**Writing – review & editing:** Floris Foijer.

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
