## [Decision Letter · Decision Letter 0]

27 Jan 2022

PONE-D-21-40038The Impact of Monosomies, Trisomies and Segmental Aneuploidies on Chromosomal stabilityPLOS ONE

Dear Dr. Raijmakers,

Thank you for submitting your manuscript to PLOS ONE. After careful consideration, we feel that it has merit but does not fully meet PLOS ONE’s publication criteria as it currently stands. Therefore, we invite you to submit a revised version of the manuscript that addresses the points raised during the review process.

As you will see, all three reviewers indicate that your manuscript would be suitable for publication if you execute your proposed revision plan fully. Reviewer 3 also raised a number of additional issues, which I encourage you to address as this would strengthen your analysis and possibly allow you to draw more meaningful conclusions. 

We look forward to receiving your revised manuscript.

Kind regards,

Daniela Cimini

Academic Editor

PLOS ONE

Journal Requirements:

[We would like to thank the Medema, Rowland and Jacobs labs for helpful discussions. This study was supported by funds from the Marie Curie Initial Training Network Project PLOIDYNET (FP7-PEOPLE-2013 and 607722) granted to R.H.M and the Dutch Cancer Society (KWF- Young Investigator Grant- 12233) granted to J.A.R.. We thank the Genomics Core Facility of the Netherlands Cancer Institute for sample preparation, data acquisition, and analysis of CNV and RNA sequencing experiments.]

[Marie Curie Initial Training Network Project PLOIDYNET:Mar Soto,Rene H. Medema FP7-PEOPLE-2013; KWF Kankerbestrijding (DCS):Jonne A. Raaijmakers KWF- Young Investigator Grant- 12233]

Reviewers' comments:

Reviewer's Responses to Questions

**Comments to the Author**

1. Is the manuscript technically sound, and do the data support the conclusions?

Reviewer #1: Yes

Reviewer #2: Partly

Reviewer #3: Yes

2. Has the statistical analysis been performed appropriately and rigorously? 

Reviewer #1: Yes

Reviewer #2: Yes

Reviewer #3: Yes

3. Have the authors made all data underlying the findings in their manuscript fully available?

Reviewer #1: No

Reviewer #2: No

Reviewer #3: Yes

4. Is the manuscript presented in an intelligible fashion and written in standard English?

Reviewer #1: Yes

Reviewer #2: Yes

Reviewer #3: Yes

5. Review Comments to the Author

Reviewer #1: In this paper, the authors describe the generation and characterization of a series of RPE1-derived clones with various segmental and whole-chromosome aneuploidies. They report that these clones generally grow more slowly than the parental cell line, and the trisomic clones show elevated chromosome missegregation. I believe that the experiments are generally well-done and these results will contribute to the growing literature on the role of aneuploidy in cancer.

I believe that the authors' plan to address the previous reviews is sufficient for publication in PLOS One. I do not think that any additional major changes are warranted.

Reviewer #2: The authors addressed many of my previous comments, and provided a rebuttal for a couple of other ones. Although I don’t necessarily agree with the rebuttal of comment #2 (proteotoxic stress) and comment #6 (demonstrating generalizability in additional cell lines/tumors), I think that the authors’ response is nonetheless reasonable.

The authors also describe a revision plan, which will: (a) explore dosage compensation in their RNAseq data and compare it to recently published articles; and (b) repeat the RNAseq experiment in UMK57-treated cells. It is important that the authors execute this revision plan, as it is expected to strengthen the manuscript's conclusions. Once these experiments are performed successfully, the paper will likely be suitable for publication in PLoS ONE.

Reviewer #3: Hintzen et al. use pharmacological inhibition of MPS1 or CENP-E to generate 27 clones from a p53kd hTERT RPE-1 cell line with various aneuploidies, which they divide into clones solely containing whole chromosome aneuploidies (15 clones, ranging from single whole chromosome aneuploidies to more complex karyotypes) and clones that contain segmental aneuploidies either solely or in combination with whole chromosome aneuploidies (12 clones). The authors describe population doubling times, rate of mitotic errors, and number of imbalanced (gained/lost) genes for each clone and analyze correlations between these variables. Additionally, the authors use MG132 and 17-AAG to induce proteotoxic stress and chromosomal missegregation in parental cells and to test sensitivity of the different aneuploid clones to proteosome/ Hsp90 inhibition. The authors fail to detect proteotoxic stress in either clones or control cells via measuring LC3BII, but using RNA-seq they find that PERK-mediated UPR response is elevated specifically in clones with trisomies but not those solely characterized by monosomy. Additionally, the authors report that several inflammatory response pathways (TNFa, IFNg) are upregulated in trisomic clones and not monosomic clones, which may be attributed to at least in part to elevated rates of CIN and micronuclei formation in the trisomic clones.

Overall, I believe the study provides some interesting data and observations within the context of this system, and, although limited in scope, will be of interest to some readers in the field. The authors have addressed many of the reviewer’s concerns and plan to perform an additional experiment to address several more. For example, the authors plan to investigate their RNAseq data for evidence of dosage compensation and will try correcting CIN using UMK57 to confirm the connection between CIN and transcriptomic changes. I believe these proposed changes will benefit the manuscript. In addition to these changes that are forthcoming, there are still a few issues that I think need to be addressed:

Minor issues

1. As a previous reviewer pointed out, you need a kinetochore marker in either live or fixed imaging to call something a lagging chromosome. The authors have changed the text in the figure legend (figure 1) to “lagging chromosome or chromosome fragment” and in the figures refer to this category as “lagging”. The term lagging should not be used for this category in the graph since acentric chromosome fragments cannot attach to the mitotic spindle and are not lagging. Also, while I certainly understand that work adds up when dealing with so many clones and the authors are hesitant to embark on a supplemental experiment to characterize mitotic defects more carefully, I think this would substantially add to the study. For example, WA9, WA14, WA10 and WA15 seem to have increases in the category that the authors refer to as “lagging”. However, from the current analysis this is essentially meaningless since we have no way to know if this is simply a reflection of increased DNA damage (like chromatin bridges) or actual lagging chromosomes, which have entirely different mechanism. The analysis is routinely performed by many labs, and even in slow growing clones there should be enough anaphase cells on 1-2 standard (e.g. 22x22mm) coverslip to scan through and get enough cells in an hour per clone, maybe a little more if they have to scan two coverslips. It’s not nearly as bad as the authors make it sound because you quickly scan over the cells that aren’t in anaphase and only consider those that are actively dividing.

2. It would be helpful if the authors provide example movies of the events observed to accompany the paper. This could exemplify the types of defects observed and allow the reader to evaluate the analysis and its limitations more easily. From the still images included in the rebuttal it can be difficult to tell the error type at the provided image quality and it isn’t labeled which image they think is which error.

3. In figure 4c, the authors indicate significance (**) when p > 0.05. Is this a mistake? Or was this result taken as statically significant and the authors are using a different alpha? I apologize if I missed a discussion of this in the manuscript.

4. For figure S4D, were any statistical tests done? I see error bars (please make sure to indicate what error bars represent in the figure legend for this and all figures). Results are discussed as changes (increased or decreased sensitivity to the drug), but it isn’t clear if they are statistically significant. Please indicate on the graph.

5. It isn’t clear to me how CIN is defined and used by the authors. For example, on page 7 they say that “clones that harbor at least one trisomy…in most cases induced CIN to different extents”. Is CIN defined as and increase above a certain threshold? Or as any statistically significant increase over the parental in instability? I think this needs to be defined in the manuscript. If the latter, then “most” would be incorrect since by looking at figure 1, five clones with trisomies (6, 13, 7, 8, 12, assuming 5 is classified as monosomy only) do not show an increase and five clones do (9, 14,10, 11, 15).

6. PLOS authors have the option to publish the peer review history of their article (what does this mean?). If published, this will include your full peer review and any attached files.

Reviewer #1: No

Reviewer #2: No

Reviewer #3: No

---

## [Author Response · Author response to Decision Letter 0]

16 Mar 2022

Reviewer #1: 

In this paper, the authors describe the generation and characterization of a series of RPE1-derived clones with various segmental and whole-chromosome aneuploidies. They report that these clones generally grow more slowly than the parental cell line, and the trisomic clones show elevated chromosome missegregation. I believe that the experiments are generally well-done and these results will contribute to the growing literature on the role of aneuploidy in cancer.

I believe that the authors' plan to address the previous reviews is sufficient for publication in PLOS One. I do not think that any additional major changes are warranted.

We thank the reviewer for his/her comments and for acknowledging that the proposed changes announced in our revision plan are sufficient to warrant publication of our manuscript in PLOS One. We have executed all planned experiments and have added them to our manuscript.

Reviewer #2: 

The authors addressed many of my previous comments, and provided a rebuttal for a couple of other ones. Although I don’t necessarily agree with the rebuttal of comment #2 (proteotoxic stress) and comment #6 (demonstrating generalizability in additional cell lines/tumors), I think that the authors’ response is nonetheless reasonable.

The authors also describe a revision plan, which will: (a) explore dosage compensation in their RNAseq data and compare it to recently published articles; and (b) repeat the RNAseq experiment in UMK57-treated cells. It is important that the authors execute this revision plan, as it is expected to strengthen the manuscript's conclusions. Once these experiments are performed successfully, the paper will likely be suitable for publication in PLoS ONE.

We thank the reviewer for reading our revision plan and for his/her response to our rebuttal. We have executed the planned experiments and have added this valuable data to the manuscript. Please find a description of the two major additions below. 

a) we explored dosage compensation on transcriptome level by evaluating the correlation between DNA copy number and RNA expression levels per chromosome in our clones. Although we find evidence in support of might observe minor compensation on a small subset of aneuploid chromosomes, this was not exclusive to monosomies and therefore unlikely explains why we do not detect CIN in these clones. We have added this analysis to our manuscript. 

b) we attempted to increase microtubule dynamics and thereby decrease CIN levels in our clones by using compound UMK57, as suggested by the reviewer. Although we collected evidence that the compound was active and used in the correct range (by dose-response curves and by observing a reduction of lagging chromosomes of two CIN cell lines, U2OS and SW620, at the selected dose), we found no significant decrease in lagging chromosomes in our aneuploid RPE-1 clones upon treatment with UMK57. Moreover, it is important to note that the majority of missegegrations in our clones are represented by chromatin bridges, a category that cannot be rescued by UMK57 in our clones, but also not in the selected CIN cell lines. Therefore, we concluded that the main cause for CIN is likely different in our clones compared to the selected CIN cell lines (i.e. not due to hyperstable kinetochore-microtubule attachments). Unfortunately, this hampered us to test whether the transcriptome changes we observed in the trisomic clones were directly driven by the enhanced CIN levels. Nevertheless, the transcriptome data on itself still convincingly show the presence of an inflammatory response in the trisomic clones and the experiment where we induce CIN and observe a similar response provides strong hints towards a role for CIN in this response. We now added the data on the UMK57 to the supplementary data as it still provided valuable data on the potential underlying causes for CIN in our clones that turn out to be likely manifold, which might reflect the pleiotropic causes for CIN in cancer. We have added this result and discussion to the manuscript.

Reviewer #3: Hintzen et al. use pharmacological inhibition of MPS1 or CENP-E to generate 27 clones from a p53kd hTERT RPE-1 cell line with various aneuploidies, which they divide into clones solely containing whole chromosome aneuploidies (15 clones, ranging from single whole chromosome aneuploidies to more complex karyotypes) and clones that contain segmental aneuploidies either solely or in combination with whole chromosome aneuploidies (12 clones). The authors describe population doubling times, rate of mitotic errors, and number of imbalanced (gained/lost) genes for each clone and analyze correlations between these variables. Additionally, the authors use MG132 and 17-AAG to induce proteotoxic stress and chromosomal missegregation in parental cells and to test sensitivity of the different aneuploid clones to proteosome/ Hsp90 inhibition. The authors fail to detect proteotoxic stress in either clones or control cells via measuring LC3BII, but using RNA-seq they find that PERK-mediated UPR response is elevated specifically in clones with trisomies but not those solely characterized by monosomy. Additionally, the authors report that several inflammatory response pathways (TNFa, IFNg) are upregulated in trisomic clones and not monosomic clones, which may be attributed to at least in part to elevated rates of CIN and micronuclei formation in the trisomic clones.

Overall, I believe the study provides some interesting data and observations within the context of this system, and, although limited in scope, will be of interest to some readers in the field. The authors have addressed many of the reviewer’s concerns and plan to perform an additional experiment to address several more. For example, the authors plan to investigate their RNAseq data for evidence of dosage compensation and will try correcting CIN using UMK57 to confirm the connection between CIN and transcriptomic changes. I believe these proposed changes will benefit the manuscript. In addition to these changes that are forthcoming, there are still a few issues that I think need to be addressed:

We thank the reviewer for acknowledging the significance of our findings and observations and for recognizing the improvements that already have been made. We have added the analysis of the dosage compensation and our results using UMK57 to the manuscript (also see comments to reviewer 2). In short: We treated our clones with UMK57 to see if we could decrease CIN levels. However, the treatment did not lead to a significant reduction in CIN. Unfortunately, this hampered us to test whether the transcriptome changes we observed in the trisomic clones were directly driven by the enhanced CIN levels. We now added the data on the UMK57 to the supplementary data.

Second, we explored dosage compensation as described above, but we did not observe evidence for large scale dosage compensation consistently being present in one category. We only observed minor compensation but this was not exclusive to monosomies and therefore we conclude that large scale dosage compensation unlikely explains why we do not detect CIN in these clones. We have added this analysis and discussion to our manuscript. 

Minor issues

1. As a previous reviewer pointed out, you need a kinetochore marker in either live or fixed imaging to call something a lagging chromosome. The authors have changed the text in the figure legend (figure 1) to “lagging chromosome or chromosome fragment” and in the figures refer to this category as “lagging”. The term lagging should not be used for this category in the graph since acentric chromosome fragments cannot attach to the mitotic spindle and are not lagging. Also, while I certainly understand that work adds up when dealing with so many clones and the authors are hesitant to embark on a supplemental experiment to characterize mitotic defects more carefully, I think this would substantially add to the study. For example, WA9, WA14, WA10 and WA15 seem to have increases in the category that the authors refer to as “lagging”. However, from the current analysis this is essentially meaningless since we have no way to know if this is simply a reflection of increased DNA damage (like chromatin bridges) or actual lagging chromosomes, which have entirely different mechanism. The analysis is routinely performed by many labs, and even in slow growing clones there should be enough anaphase cells on 1-2 standard (e.g. 22x22mm) coverslip to scan through and get enough cells in an hour per clone, maybe a little more if they have to scan two coverslips. It’s not nearly as bad as the authors make it sound because you quickly scan over the cells that aren’t in anaphase and only consider those that are actively dividing.

We have now performed fixed analysis of the clones WA9, WA14, WA10 and WA15. These clones were selected as they displayed an increased level of ‘lagging’ chromosomes or fragments as judged by live cell imaging. We now changed the category lagging as judges by live cell imaging, to ‘lagging DNA’ throughout the manuscript to avoid confusion. In anaphase both categories (centric intact chromosomes or acentric fragments will “lag” behind the main chromatin pack and will therefore appear as lagging DNA). We thereafter performed the proposed fixed experiments to categorize the lagging DNA category into centric and acentric fragments. For this, we stained the cells with CREST serum to visualize centromeres combined with DAPI and phospho-Histone H3 (ser10) to identify mitotic cells. We found that the fixed data nicely resembled the data we obtained with our live-cell imaging in terms of the number of cells displaying chromosome segregation errors. We confirmed that the largest category of missegregations amongst all clones were represented by chromatin bridges. Moreover, we found that the fraction of lagging DNA with and without centromeres differed per clone, but overall the category that was observed most was chromosomes containing a centromere. This points towards mitotic defects underlying the majority of lagging chromosomes. Lagging chromosomes can be caused for example by defect in KT-MT attachments (enhanced merotelics). Importantly, such defects can also be induced indirectly by for example low levels of replication stress (Wilhelm et al. 2019 Nat. Comm.). In conclusion of the experiments we performed with UMK-57, we conclude that hyper-stable KT-MT attachments are unlikely underlying the observed lagging chromosomes. We have added this data to our manuscript and discussed these results in light of previous literature and the additional findings we presented in this manuscript. 

2. It would be helpful if the authors provide example movies of the events observed to accompany the paper. This could exemplify the types of defects observed and allow the reader to evaluate the analysis and its limitations more easily. From the still images included in the rebuttal it can be difficult to tell the error type at the provided image quality and it isn’t labeled which image they think is which error.

We thank the reviewer for this suggestion and have added example movies of the different error types. 

3. In figure 4c, the authors indicate significance (**) when p > 0.05. Is this a mistake? Or was this result taken as statically significant and the authors are using a different alpha? I apologize if I missed a discussion of this in the manuscript.

We apologize for this mistake. We went back to the data and found that we made a typo and left out one 0. The p-value of 0.056 should be 0.0056. We have corrected this mistake. 

4. For figure S4D, were any statistical tests done? I see error bars (please make sure to indicate what error bars represent in the figure legend for this and all figures). Results are discussed as changes (increased or decreased sensitivity to the drug), but it isn’t clear if they are statistically significant. Please indicate on the graph.

We performed an ordinary one-way ANOVA between parental and each clone and have added this to the graph. Clone 14.20, which displays the largest difference from parental is the only clone that is significantly different ( p-value = 0.0097). We have added this information to the figure legends and to the main text. 

5. It isn’t clear to me how CIN is defined and used by the authors. For example, on page 7 they say that “clones that harbor at least one trisomy…in most cases induced CIN to different extents”. Is CIN defined as and increase above a certain threshold? Or as any statistically significant increase over the parental in instability? I think this needs to be defined in the manuscript. If the latter, then “most” would be incorrect since by looking at figure 1, five clones with trisomies (6, 13, 7, 8, 12, assuming 5 is classified as monosomy only) do not show an increase and five clones do (9, 14,10, 11, 15).

We agree with the reviewer that our definition of CIN was not entirely clear. We now stated a clearer definition of CIN as any level of missegregations that exceeds the basal level present in parental cells and control clones (i.e. above 10%). We have added this definition to our manuscript for clarification.

---

## [Decision Letter · Decision Letter 1]

13 Apr 2022

PONE-D-21-40038R1The Impact of Monosomies, Trisomies and Segmental Aneuploidies on Chromosomal stabilityPLOS ONE

Dear Dr. Raaijmakers,

Thank you for submitting your manuscript to PLOS ONE. After careful consideration, we feel that it has merit but does not fully meet PLOS ONE’s publication criteria as it currently stands. Therefore, we invite you to submit a revised version of the manuscript that addresses the points raised during the review process.

As you will see, the reviewers felt that your revisions have greatly improved your manuscript. However, reviewer 2 has some additional concerns. We believe that addressing those remaining issues will be important. The issues are minor and we hope you will be able to address them in a short time frame,

We look forward to receiving your revised manuscript.

Kind regards,

Daniela Cimini

Academic Editor

PLOS ONE

Journal Requirements:

Reviewers' comments:

Reviewer's Responses to Questions

**Comments to the Author**

1. If the authors have adequately addressed your comments raised in a previous round of review and you feel that this manuscript is now acceptable for publication, you may indicate that here to bypass the “Comments to the Author” section, enter your conflict of interest statement in the “Confidential to Editor” section, and submit your "Accept" recommendation.

Reviewer #2: (No Response)

Reviewer #3: All comments have been addressed

2. Is the manuscript technically sound, and do the data support the conclusions?

Reviewer #2: Partly

Reviewer #3: Yes

3. Has the statistical analysis been performed appropriately and rigorously? 

Reviewer #2: Yes

Reviewer #3: Yes

4. Have the authors made all data underlying the findings in their manuscript fully available?

Reviewer #2: Yes

Reviewer #3: Yes

5. Is the manuscript presented in an intelligible fashion and written in standard English?

Reviewer #2: Yes

Reviewer #3: Yes

6. Review Comments to the Author

Reviewer #2: The authors made some effort but were unable to successfully address the two issues that were highlighted by myself and by Reviewer #3 in the previous round of comments:

(1) The dosage compensation analysis that is shown in the Discussion and in Supplementary Fig. 7 is partial and incomplete. A pathway-level compensation analysis should be performed, and so is a more direct comparison to the published literature on that matter.

(2) The UMK57 experiment failed. While the authors provide a possible explanation for this failure, it still leaves us without a direct connection between CIN and the observed transcriptomic changes. So it is still impossible to determine which of the observed gene expression changes are due to chromosome imbalance and which are due to CIN. This should be clearly stated in the Discussion.

Reviewer #3: The authors have addressed my previous comments and carried out additional analysis to differentiate mitotic defects based on whether the DNA that does not segregate with the main chromosome mass is CREST+ or acentric. The authors also carried out their proposed research plan. While those results were not conclusive/ positive results, I believe the changes overall strengthen the manuscript and, in my opinion, the manuscript with the current changes is acceptable for publication in PLoS One.

7. PLOS authors have the option to publish the peer review history of their article (what does this mean?). If published, this will include your full peer review and any attached files.

Reviewer #2: No

Reviewer #3: No

---

## [Author Response · Author response to Decision Letter 1]

29 Apr 2022

Response to reviewers

Reviewer #2: 

The authors made some effort but were unable to successfully address the two issues that were highlighted by myself and by Reviewer #3 in the previous round of comments:

(1) The dosage compensation analysis that is shown in the Discussion and in Supplementary Fig. 7 is partial and incomplete. A pathway-level compensation analysis should be performed, and so is a more direct comparison to the published literature on that matter.

In the preprint by Schukken et al., they investigated dosage compensation both on RNA level as well as protein level using a very large data set containing 367 cell lines from the Cancer Cell Line Encyclopedia (CCLE). They found that mean mRNA and protein expression levels increases upon arm gain and decreases upon arm loss. However, on the individual gene level, they observed that a large subset of genes was ‘buffered’. However, it is very important to note that this observation mostly applied to buffering at the protein level as the majority of RNA transcripts scaled with the DNA copy number (in line with previous studies, see Fehrmann et al. 2015). The Schukken study performed GO and pathway enrichment for the genes that were found to be buffered. When performing this analysis for proteins, they found enrichment for ribosomal genes, RNA processing genes and protein complex genes. These findings nicely reflect previously published results for chromosome gains (Stingele et al. 2012, Dephoure et al. 2014, Brennan et al. 2019). Importantly, Schukken et al. found similar Gene Ontologies and pathways to be buffered for gains and losses. A rather different result was obtained when performing a similar analysis for buffered RNA’s: first, they only found three pathways to be enriched, which were only marginally significant. Second, these pathways were different between gains and losses and did not seem to involve expected pathways. Therefore, the authors concluded that: “This lack of dosage compensation at the RNA level indicates that translational or post-translational regulation, rather than transcriptional regulation, drives the pervasive dosage compensation of buffered proteins and protein complex subunits that we have observed in aneuploid cancer cells.”. We completely agree with this line of reasoning and therefore, we feel that performing pathway enrichment on our transcriptome data would not lead to any relevant insight.

In summary, although we could try to perform pathway enrichment analysis on our data set, we believe this will not result in meaningful data for the following reasons:

1. Our data set is way too small to reliably assign genes to be buffered or not. RNA sequencing data is intrinsically noisy and for most aneuploidies we only have one clone with that specific aneuploidy. Therefore, we are not able to reliably classify a gene as buffered or not with sufficient confidence.

2. Most buffering occurs on protein level, as also found by Schukken et al. and as suggested by many other publications such as by Gonçalves et al. 2017. Since we only obtained RNA data, these findings will unlikely result in any valuable conclusions.

Thus, if we would do such analysis, it would at most lead to insignificant data that rather than leading to pushing the field forward, it could lead to confusion in the field. To address the questions regarding dosage compensation on transcriptome level on the individual gene level/pathway level, a much larger data set is required, preferentially covering all chromosomes multiple times. Instead, we propose to highlight the limitations of our data set in the manuscript and make clear that larger datasets are required to make solid conclusions on compensatory mechanisms on the transcriptome and proteome level. We discussed this with the scientific editor and we agreed to clarify these limitations in the discussion.

In the discussion we state the following:

“However, we need to note that the amount of data is limited and that there is also noise observed around the chromosomes that are euploid. Also, we lack the resolution to conclude if there is dosage compensation on the level of individual, possibly critical, genes. Finally, and most importantly, to make solid conclusions on dosage compensation, we require proteome data as we assume the biggest consequences of dosage compensation come from mechanisms associated with protein degradation.” 

(2) The UMK57 experiment failed. While the authors provide a possible explanation for this failure, it still leaves us without a direct connection between CIN and the observed transcriptomic changes. So it is still impossible to determine which of the observed gene expression changes are due to chromosome imbalance and which are due to CIN. This should be clearly stated in the Discussion.

We agree with the reviewer that we did not clearly mention in the discussion that we were unable to directly investigate the relationship between CIN and the observed transcriptomic changes. We therefore added this to the discussion: “However, since we were not able to rescue CIN in our clones using UMK57, we can only speculate which transcriptomic changes are driven by CIN and which are caused by chromosome imbalances. Future work aiming to understand the underlying mechanisms of CIN in our clones could open up new strategies to reduce CIN and to directly test this relationship.”

Reviewer #3: 

The authors have addressed my previous comments and carried out additional analysis to differentiate mitotic defects based on whether the DNA that does not segregate with the main chromosome mass is CREST+ or acentric. The authors also carried out their proposed research plan. While those results were not conclusive/ positive results, I believe the changes overall strengthen the manuscript and, in my opinion, the manuscript with the current changes is acceptable for publication in PLoS One.

We thank the reviewer for his/her support for publication in PloS One.

---

## [Editor Report · Decision Letter 2]

3 May 2022

The Impact of Monosomies, Trisomies and Segmental Aneuploidies on Chromosomal stability

PONE-D-21-40038R2

Dear Dr. Raaijmakers,

We’re pleased to inform you that your manuscript has been judged scientifically suitable for publication and will be formally accepted for publication once it meets all outstanding technical requirements.

Kind regards,

Daniela Cimini

Academic Editor

PLOS ONE
---

## [Editor Report · Acceptance letter]

22 Jun 2022

PONE-D-21-40038R2 

The Impact of Monosomies, Trisomies and Segmental aneuploidies on Chromosomal stability 

Dear Dr. Raaijmakers:

I'm pleased to inform you that your manuscript has been deemed suitable for publication in PLOS ONE. Congratulations! Your manuscript is now with our production department. 

Kind regards, 

on behalf of

Dr. Daniela Cimini 

Academic Editor

PLOS ONE